



# High-Fidelity Processing of Instantaneous Line-of-Sight Returns from Nacelle-Mounted Lidar including Supervised Machine Learning

Kenneth A. Brown[1], Thomas G. Herges[1]

[1]Sandia National Laboratories, Albuquerque, 87123, U.S.A.

*Correspondence to*: Kenneth A. Brown (kbrown1@sandia.gov)

**Abstract.** Wind turbine applications that leverage nacelle-mounted Doppler lidar are hampered by several sources of uncertainty in the lidar measurement, affecting both bias and random error. Two problems encountered especially for nacelle-mounted lidar are solid interference due to intersection of the line of sight with solid objects behind, within, or in front of the measurement volume, as well as spectral noise due primarily to limited photon capture. These two uncertainties can be reduced

with high-fidelity quality assurance/quality control (QA/QC) processing techniques. Our work compares three QA/QC techniques, including conventional thresholding, advanced filtering, and a novel application of supervised machine learning with ensemble neural networks, based on their ability to reduce uncertainty introduced by the two observed non-ideal spectral features. The approach leverages data from a field experiment involving a continuous-wave (CW) SpinnerLidar from the Technical University of Denmark (DTU) that provided scans of a wide range of flows both unwaked and waked by a field

turbine. Independent measurements from an overlapped meteorological tower permit experimental validation of the instantaneous velocity uncertainty remaining after QA/QC processing that stems from solid interference and strong spectral noise, which is a validation that has not been performed previously. All three methods perform similarly for non-interfered returns, but the advanced filtering and machine learning techniques perform better when solid interference is present, which allows them to produce overall standard deviations of error between 0.2 and 0.3 m/s, or a 1-22% improvement versus the

conventional thresholding technique, over the rotor height for the unwaked cases. Between the two improved techniques, the advanced filtering produces 3.5% higher overall data availability, while the machine learning offers a faster runtime (i.e, ~1 second to evaluate) that is therefore more commensurate with the requirements of real-time turbine control. The QA/QC techniques are described in terms of application to CW lidar, though they are also relevant to pulsed lidar. Previous work by the authors (Brown and Herges, 2020) explored a novel attempt to quantify uncertainty after a lidar QA/QC process using

simulated lidar returns; this article provides true uncertainty quantification versus independent measurement and does so for three rather than one QA/QC techniques.

## 1. Introduction

Despite the continuing growth of wind energy technology, several sub-fields of wind energy are still not mature (Veers et al., 2019). Real-time control of turbines within the stochastic atmosphere and better understanding of turbine-to-turbine wake





interactions represent two areas needing further advances and areas for which accurate wind-field sensing around the turbine
is imperative. Such sensing is enabled through Doppler lidar instruments, and nacelle-mounted lidar, in particular, have made
recent inroads with applications in monitoring and control (Harris et al., 2006; Mikkelsen et al., 2013; Simley et al., 2014;
Simley et al., 2018) and model validation (Doubrawa et al., 2020; Brown et al., 2020; Hsieh, 2021). Continuing investment in
such lidar technology includes efforts to reduce the uncertainty of wind field measurements over the whole field of view, which

is critical for both forward-mounted lidar used in feedforward control applications and rear-mounted lidar used in wake
measurements for model validation. Uncertainties in lidar measurements stem both from the lidar line-of-sight velocity, $u^{los}$,
readings themselves and from inaccuracies in modeling approaches for reconstruction of the velocity vector (Lindelöw-
Marsden, 2009; Van Dooren, 2021). This work focuses on quantification of the former, more fundamental source of lidar
uncertainty that is present in all lidar measurements regardless of any flow reconstruction approach that is later applied to the

data.

An example of the raw return from a line-of-sight reading is given in Figure 1. The fast-Fourier-transformed power spectral
density, $s$, returned from the scattering along the laser path is distributed across a range of Doppler shift frequencies, $f$, which
are related to line-of-sight velocity according to $u^{los} = \lambda f / 2$, where $\lambda$ is the wavelength of the laser. Some aspects of the uncertainty in $u^{los}$ have been found to be small for typical setups such as the accuracy of the positioning of the line of sight itself (Herges et al., 2017) and beam motion during data capture (i.e., the *blurring effect*) (Simley et al., 2014). Other aspects are well documented and can be quantified *a priori* through virtual lidar techniques. Most notably, there is significant broadening of the lidar spectra (and thus alteration of the mean and fluctuating components of the time series of $u^{los}$) from flow inhomogeneities such as mean gradients and turbulence within the measurement volume (Stawiarski et al., 2013; Simley et al., 2014; Wang et al., 2016; Forsting et al., 2017; Sekar et al., 2020). This broadening, which is also a function of the line-of-sight weighting distribution for a CW lidar, is observed as the

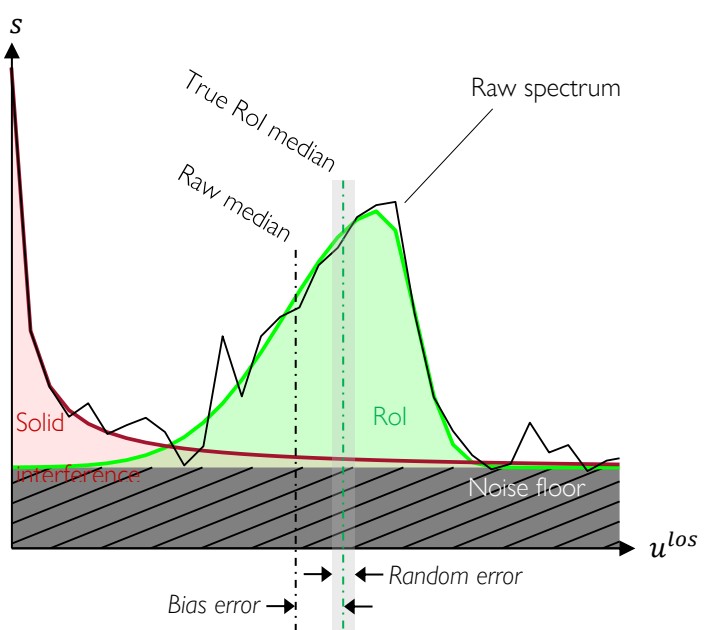

**Figure 1. Example power spectral density, $s$, distribution versus line-of-sight velocity, $u^{los}$, of a raw lidar return illustrating the contamination of the region of interest (RoI) by solid interference and amplitude noise. The raw geometric median contains bias error due to the solid interference as well as random error due to the amplitude noise. Figure adapted from Brown and Herges (2020).**



width of the region of interest (RoI) in Figure 1. On the other hand, we find several error sources in measured lidar results
whose impact cannot be known *a priori*. These sources are due to spectral features embedded in the lidar signal that stem
both from instrument error and from non-aerosol returns as shown in Figure 1, and these are especially prevalent for nacelle-
mounted lidar as described below.

Amplitude noise in the spectrum, which is depicted by the localized peaks as well as the presence of a flat noise floor
(Harris et al., 2006) in Figure 1, results in a loss of precision (i.e., larger spread from the true value) in the velocity estimation
from the RoI. The intensity of the noise, which for modern lidar is due primarily to shot noise (Peña and Bay Hasager, 2013),
depends on the range-resolved intensity of the backscatter (Liu et al., 2006). Therefore, appropriate shot-noise error analyses
should account for the unique noise content observed in each lidar return, which cannot be determined *a priori* (Simley et al.,
2014). A particular configuration of interest to our work is fast-scanning (i.e., ~500 Hz) CW lidar as have been mounted on
turbine nacelles (Mikkelsen et al., 2013). One drawback of this configuration, however, is that the high temporal resolution
trades with shorter averaging times that yield higher instrument error due to low carrier to noise ratio, *CNR* (Angelou et al.,
2012).

Interference from solid surfaces that intersect the probe volume introduces a source of bias in lidar readings, and the severity
of the interference again cannot be determined *a priori*. For nacelle-mounted lidar, such interference is common and stems
from the surrounding terrain, optical windows (i.e., boresight interference, which occurs near the center of the field of view
(Brown and Herges, 2020)), neighboring turbines, meteorological towers, and the turbine's own blades for the case of a
forward-facing lidar mounted on the top of the nacelle (rather than on the spinner). A characteristic spike shape in the Doppler
return near zero velocity indicates the presence of solid interference.

The first tactic against spectral noise and solid interference is QA/QC processing. In the context of non-stationary
atmospheric measurements, the most simplistic QA/QC approach is thresholding (Angelou et al., 2012; Peña and Bay Hasager,
2013) where spectral bins with $s$ magnitudes less than a specified threshold are nulled (if no signal remains below this
threshold, then the lidar data can be rejected (Frehlich, 1996)). Signal strength can be increased by accumulating the spectra
over multiple laser pulses along a single line of sight (Rye and Hardesty, 1993), and this approach has become mainstream in
both pulsed and CW lidar technologies. Conversely, if the spectrum peak magnitude is too high, the data may be rejected on
the grounds that a solid return has been captured. Rather than reject such a solid return outright, Godwin et al. (2012) worked
to mitigate ground interference bias for airborne pulsed lidar, though their approach was admittedly subject to a large degree
of subjectivity in defining certain thresholds and was also unable to handle wind speeds near the interference velocity. Herges
and Keyantuo (2019) developed another technique that employs a user-defined set of filters to carefully estimate the bounds
of the RoI, thus removing the impact of features due to solid interference and spectral noise away from the RoI.

The next step in lidar processing is the mean frequency estimation (i.e., estimation of the Doppler frequency shift, which
yields the line-of-sight velocity estimate). Mean frequency estimators (MFEs) have long been studied for radar and lidar
applications including recent work with parameter estimation on spectra from fast-Fourier-transformed signals. Specific to
pulsed lidar measurements, Lombard et al. (2016) examined five such estimators including the maximum, centroid, matched



filter, maximum likelihood, and polynomial fit MFEs and found that all estimators save the first offer suitable accuracy compared to the theoretical ideal performance of the Cramer-Rao lower bound. Specific to CW lidar measurements, Held and
Mann (2018) examined the maximum, centroid, and median MFEs and found the highest accuracy when validating lidar results against sonic anemometer measurements for the median MFE followed by the centroid and finally maximum MFEs. Thus, the median MFE has become the most common estimator used in wind energy.

After the frequency estimation, another layer of QA/QC can be applied through despiking techniques that reject outliers in a time series, such as the classical standard deviation filter, iterative standard deviation filter (Hojstrup, 1993; Vickers and
Mahrt, 1997; Newman et al., 2016), or inter-quartile range filter (Hoaglin et al., 1984; Wang et al., 2015). Leveraging assumptions related specifically to lidar configurations, Forsting and Troldborg (2016) describe a finite-difference-based despiking technique that importantly considers spatial as well as temporal gradients. Beck and Kühn (2017) introduced an adaptive filtering technique, though it relies on the assumption of self-similar flow over a span of time.

Above we have reviewed how the type of QA/QC processing as well as mean frequency estimation have bearing on the
accuracy of the final quantities of interest (QoIs). All the described techniques work to mitigate amplitude noise and solid interference but involve a degree of subjectivity in defining certain processing parameters. While appropriate selections of these parameters can be found for specific conditions and Doppler return shapes, there does not exist a universal set of optimal parameters even for the selection of the simple noise threshold (Angelou et al., 2012; Gryning et al., 2016), which makes the application of the de-noising techniques prone to over- or under-estimating QoIs. Working towards a solution, Brown and
Herges (2020) quantified residual uncertainty in QoIs after the QA/QC process by processing synthetic spectra with known ground truth properties that mimicked the shape of measured spectra.

The ultimate test of the accuracy of the QoI estimation, however, is experimental validation against an independent measurement. In terms of validation of the uncertainty of lidar techniques, most work has been performed on time-averaged samples, typically over a 10-minute window as specified in the industry standard for power performance assessment
(Commission, 2005). For instance, Smith et al. (2006), Albers et al. (2009), Slinger and Harris (2012), Gottschall et al. (2012), Hasager et al. (2013), Wagner and Bejdic (2014), Giyanani et al. (2015), and Cariou et al. (2013) all compare lidar-derived velocities to traditional anemometer-derived velocities over 10-minute bins, often returning regression slopes and coefficients of determination within 0.01 of unity.

For turbine control and model validation purposes, however, the uncertainty of interest is the instantaneous one, for which
values are significantly larger and the volume of previous work is significantly smaller. Courtney et al. (2008) reported instantaneous errors between co-located lidar probe volumes and cup anemometers to have standard deviation of 0.2 m/s and mean bias between -0.2 to 0.2 m/s, though they noted that the actual values depend on the distribution of wind speeds. A wind tunnel experiment by Van Dooren et al. (2021) showed instantaneous velocity from a co-located lidar probe volume and hotwire anemometer with coefficients of determination much smaller than the 10-minute-averaged results above (i.e., $0.65 <$
$R^2 < 0.95$). As Pedersen and Courtney (2021), for instance, have shown that the standard error in line-of-sight velocity measured versus a hard target for a CW lidar is on the order of 0.1%, the main source of errors observed by Courtney et al.





(2008) and Van Dooren et al. (2021) is understood to be flow inhomogeneity and amplitude noise (neither of these cases included solid interference effects).

Like these last studies, this article considers instantaneous data from CW lidars in the face of flow inhomogeneity and 135 amplitude noise. In contrast to the previous work, our work explicitly compares the uncertainty of several end-to-end QA/QC techniques and does so over a wider range of flows and lidar return types than has been done previously. Specifically, we examine flows that are both unwaked and waked by a field turbine, including those where the specific nacelle-mounted lidar problems of solid interference and amplitude noise present a particular challenge. The objective is to bound the achieved uncertainty in each of the QA/QC processes for the most common QoI: the spectral (i.e., geometric) median line-of-sight 140 velocity, $\tilde{u}^{los}$. To evaluate the efficacy of the interference and noise rejection processes, we compare $\tilde{u}^{los}$ to corresponding values measured from a meteorological tower co-located with the lidar focus point.

This work is novel not only by nature of the strides taken to quantitatively determine instantaneous lidar QoI uncertainties but also in the first-time exploration, benchmarking, and stress-testing of two high-fidelity processing techniques. The study compares the accuracy of $\tilde{u}^{los}$ as processed from measured lidar spectra in three parallel ways: (1) with the conventional 145 thresholding technique, (2) with the advanced filtering technique of Herges and Keyantuo (2019), and (3) with a novel application of an ensemble machine learning model that is trained on spectral data mimicking those observed in the field.

In the remainder of the article, the methodology underlying the three processing techniques is outlined in Section 2, followed by an overview of the demonstration experiment in Section 3, validation results in Section 4, discussion in Section 5, and concluding remarks with future work in Section 6.

**2. Processing Techniques**

The three techniques for QA/QC processing of nacelle-mounted lidar data are described in this section.

**2.1. Thresholding**

The thresholding technique used herein is related to the conventional thresholding approaches given in Angelou et al. (2012) and Peña and Bay Hasager (2013). First, a check is made on the magnitude of the first useable velocity bin of the spectrum, 155 and the return is rejected if this magnitude is the maximum among the bins, which often occurs when dominant solid interference is present. Otherwise, the mean noise level in the spectrum, $\mu_{noise}$, and standard deviation of noise, $\sigma_{noise}$, are calculated over the last 100 bins of each spectrum similar to Simley et al. (2014). These last 100 bins are in the tail of the spectrum sufficiently away from the RoI (beyond the right edge of Figure 1). The thresholded power spectrum, $s_{th.}$, is then calculated from the raw spectrum, $s$, via Eq. (1):



$$s_{th.} = s - \mu_{noise} - n_\sigma \sigma_{noise}, \tag{1}$$

where $n_\sigma$ is a tunable parameter for the number of $\sigma_{noise}$ above the noise floor of the desired threshold level. Negative values

of $s_{th.}$ are subsequently set to zero, and the spectral median is calculated according to standard practice as embodied in the

MATLAB function `medfreq`, which defines the median frequency as that which divides the spectrum into two equal areas.

      As thresholding requires a degree of data loss, there exists a tradeoff between reduction in random error in a thresholded

timeseries due to rejection of spurious spectral noise and increase in random and bias errors due to reduced $CNR$ and altered

skew of the distribution, respectively. The optimal value of $n_\sigma$ for CW lidar depends at least on the spectral width as described

in Angelou et al. (2012), and we choose an $n_\sigma$ of five so that any signal above this threshold can conservatively be regarded

as from the wind rather than from noise (Peña and Bay Hasager, 2013). In this case, we do not apply any correction for the

alteration of the skew distribution due to thresholding.

### 2.2. Advanced Filtering

The advanced filtering technique described by Herges and Keyantuo (2019) and also implemented in this article builds on the

thresholding technique to maintain a greater data availability while reducing both random and bias errors. The technique

leverages the lidar spectral data throughout an entire scan (i.e., incorporating information from adjacent scan positions) to

isolate the velocity field of interest within the spectra, remove signals from solid returns, and reduce noise using a bilateral

filter. The advanced filtering technique was developed by matching known erroneous measurements within the lidar scan

rosette to patterns determined from feature identification within the Doppler spectral image, which includes the spectral

information throughout the entire scan. The feature identification within the spectral image is used to identify and remove hard

targets, low-signal returns, and returns from nearby non-stationary wind turbine blades. An additional outlier detection was

developed as a two-dimensional implementation of traditional despiking methods to catch remaining outliers within the scan

pattern. An overview of the technique is provided below while Herges and Keyantuo (2019) explain the advanced filtering

technique in greater detail.

      The single 2-second example rosette scan with 984 points, or scan indices, shown in Figure 2(a), was chosen to describe

how the advanced filtering method works, and this method holds for all DTU SpinnerLidar data collected at the SWiFT site,

including data with inflow variations in wind speed, turbulence, shear, veer, and aerosol particulate concentrations, throughout

all focus distances and scan-head motor speed rates. This example scan, which was taken with the lidar and turbine in a

different orientation than for the rest of the article, was measured with a focus distance of $5D$, or 135 m, in the direction of a

downstream turbine (i.e., WTGa2 in Figure 7) that was operating within the lidar field of view and a wake from the lidar's

own turbine (i.e., WTGa1).





Figure 2 shows the unprocessed input to the advanced filtering technique. The figure includes the rosette scan pattern in (a), seven example Doppler spectral traces versus $u^{los}$ in (b), and the spectral image created by the inclusion of all normalized

power spectral density distributions concatenated in time along the scan index as shown in (c). The advanced filtering technique primarily uses the data format of the spectral image in Figure 2(c), with Figures 2(a) and (b) included to help interpret the information contained in the image. Line colors correspond between each of the three subfigures and indicate seven scan indices of interest; the indices were chosen to help demonstrate the advanced filtering technique across a wide subset of return types (i.e., even wider than will be considered in the later sections of this paper). As seen in Figure 2(a), the indices of interest

include example returns from each of the following: the undisturbed freestream of the atmospheric boundary layer, the center of the wake, the edge of the wake, the boresight, the ground, and the rotating downstream rotor. Note that Figure 2(a) also shows an outline of WTGa2 and the wake from WTGa1 (in red) to give reference of what was physically occurring at the locations in the scan relative to the seven example locations.

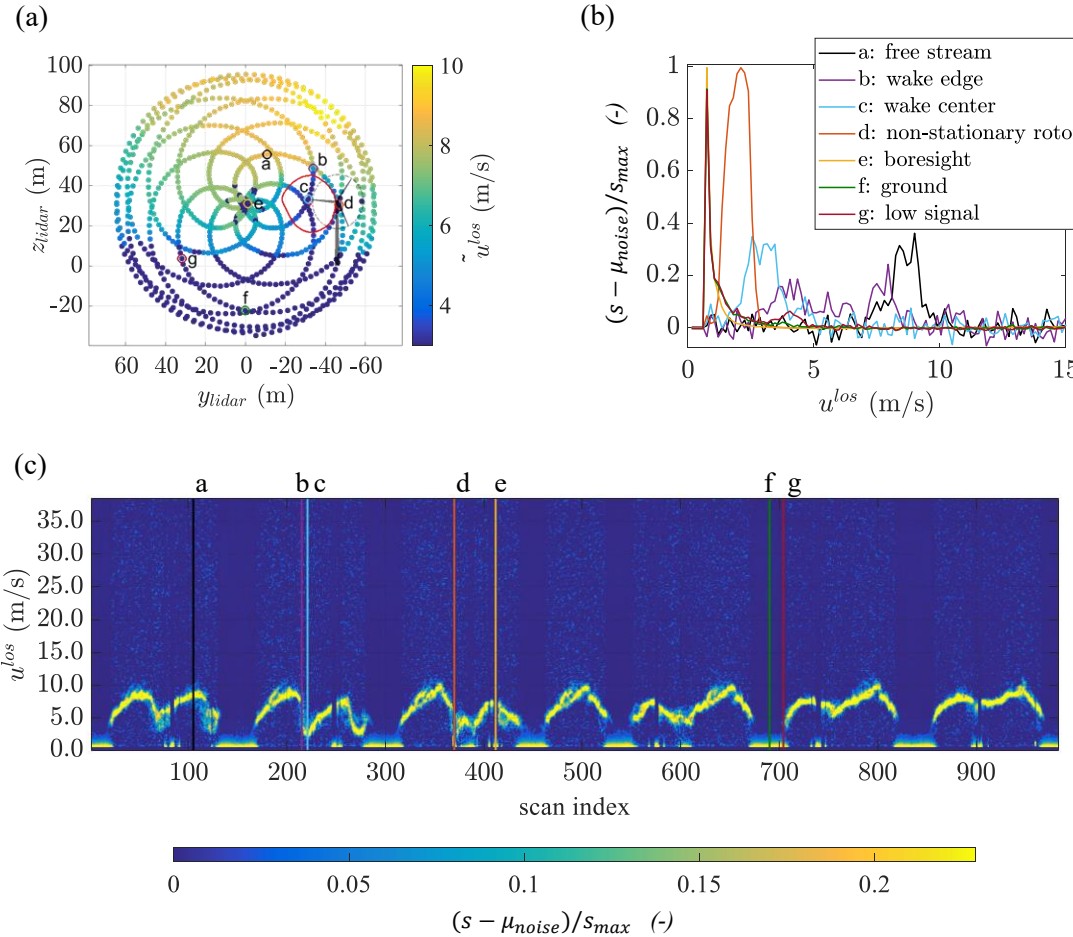

**Figure 2. Unprocessed Doppler spectra shown in three forms with tracking of seven example cases (indicated by the letters a.-g.): (a) colormap of spectral median line-of-sight velocity, $\widetilde{u}^{los}$, over the rosette scan pattern including the wake (as indicated by the red line and center dot) and relative location to the downstream turbine; (b) normalized power spectral density, $(s - \mu_{noise})/s_{max}$, versus $u^{los}$ where $s_{max} = 2^{16} - 1$; and (c) Doppler spectral image of $(s - \mu_{noise})/s_{max}$. In (a), $y_{lidar}$ and $z_{lidar}$ are the lateral and vertical coordinates, respectively, of the lidar coordinate system.**

Figure 3 displays the effects of the primary steps of the advanced filtering technique on the seven example $u^{los}$ traces. The first step (i.e., moving between Figure 2(b) and Figure 3(a)) was to remove the effects of the solid returns within the spectral image using a mask. The mask was created by proportionally projecting the signal strength at the lowest velocity bin into higher velocity bins. The values for the linear projection were determined empirically and may be specific to a given lidar device. However, the values held for the SpinnerLidar in this experiment throughout all focus distances. The mask regions




were then increased horizontally by 2 pixels using the image processing technique of morphological dilation to ensure the regions fully masked the effect of the solid returns. The regions within the spectral image covered by the mask were zeroed out. Note that the high-strength ground return portion of the "low signal" example was removed while the low-strength portion, returned from the aerosols, remains.

    The next step in the technique (i.e., moving between Figure 3(a) and (b)) includes a combination of filtering to remove shot

noise, thresholding, and identification of the RoI, the latter of which includes flow information from both the atmospheric boundary layer and wake. A weak bilateral filter[1], the effect of which can be observed by comparing the noise in the wake edge distribution between Figure 3(a) and (b), is believed to be more effective and accurate at reducing shot noise, as compared to a one-dimensional filter, because it utilizes the Doppler information from surrounding measurement points within the continuous flowfield. The RoI within the spectral image was created by preserving the Doppler spectra above the normalized

threshold of 0.015. Smaller regions of spectra outside the RoI remained because of noise values above the threshold or signal returns from rotating blades, and these regions were removed if they were not interconnected with the primary flowfield RoI,

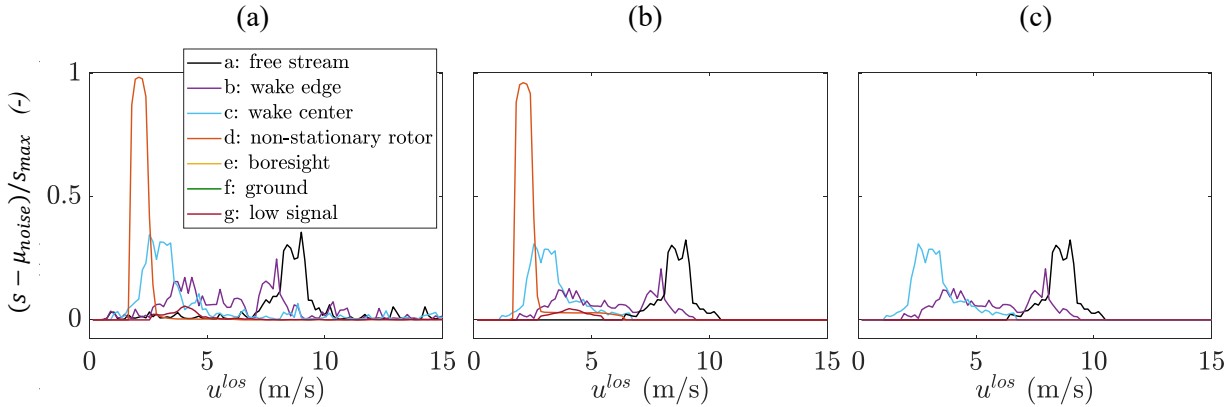

**Figure 3. Traces of normalized power spectral density, $(s - \mu_{noise})/s_{max}$, versus $u^{los}$ where $s_{max} = 2^{16} - 1$. These cases demonstrate the progression of the primary advanced filtering steps: (a) removal of hard targets; (b) bilateral filtering, thresholding, and isolation of ROI; and (c) combined outlier detection method and low signal quality filter.**

which is determined as the large region that intersects the expected $\tilde{u}^{los}$ values. The thresholding value was chosen empirically as a value that approaches zero without including regions of noise interconnected with the primary RoI. Remaining invalid measurements from cases that are interconnected with the flowfield RoI and have a low $CNR$ or are interconnected with the

rotating downstream rotor blades were addressed in subsequent steps.

---

[1] A bilateral filter is an edge preserving nonlinear filter that replaces the intensity of each pixel with a Gaussian weighted average of intensity values from nearby pixels, which is a common filtering technique for reducing shot noise within images (Phelippeau et al., 2008).



Two additional filters were used in the final step (i.e, moving between Figure 3(b) and (c)) to remove the remaining invalid measurements. The first filter is a combination of two outlier detection methods (Figure 4 (a) and (b)), and the second filter removes returns with low signal quality (Figure 4(c)). The two initial outlier detection methods are used to capture the returns from nonstationary solid targets when the flowfield among neighboring scan indices has similar velocities. The first outlier detection method uses a spatially smoothed scan pattern, $\tilde{u}_{smooth}^{los}$, that is a time-weighted average of $\tilde{u}^{los}$, calculated from the spectral median of the $u^{los}$ traces in the filtered spectral image, within a sliding neighborhood to detect outliers from the difference between the spatially-smoothed scan and the unsmoothed pattern $(\tilde{u}^{los} - \tilde{u}_{smooth}^{los})$ as in Figure 4(a). The filtered spectral image is the collection of all $u^{los}$ traces concatenated in time and processed to the step corresponding to the example traces in Figure 3(b). Figure 4(a) shows the size of the sliding neighborhood as a black circle, and there are two invalid measurements observable near the rotating downstream rotor. The invalid measurements were identified using a threshold above six standard deviations of $\tilde{u}^{los} - \tilde{u}_{smooth}^{los}$ across all 984 scan indices. However, this velocity metric was not robust enough to work for all scenarios of data captured during extended field testing without including the second outlier detection step. Figure 4(b) shows the normalized power spectral density peak prominence, $(s_{peak} - \mu_{noise})/s_{max}$, of the filtered spectral image at each scan point. Again, the peak return signals from the operational rotor are clear. A threshold of the mean $(s_{peak} - \mu_{noise})/s_{max}$ across all 984 scan indices plus four standard deviations detected peak signal outliers. The velocity difference smoothing and signal peak outlier detections were combined to robustly capture the effect of the operational rotor at all focus distances, removing only data that qualified as outliers using both detection methods and thus effectively removing the erroneous non-stationary solid targets. The second filter applied during the final step removes power spectra distributions with low signal quality. This filter uses the reciprocal of a signal quality metric of the filtered spectral image. The reciprocal metric is $s_{max}/(s_{peak} - \eta_{noise})$, where $\eta_{noise}$ is the median value of the noise peaks over a region of $u^{los}$ similar as that used to calculate $\mu_{noise}$ (i.e., away from the RoI). Figure 4(c) displays the reciprocal quality metric and removes values above 20, which correspond to a peak to noise difference of less than 5%. This filter removes the bright values in the ground region of Figure 4(c) and also removes values from scans with periods of reduced aerosol within the atmospheric boundary layer, which are not observed in this example scan.

Figure 3 (c) shows the final result of the example traces using the advanced filtering method, leaving only the RoIs in the freestream, wake center, and wake edge cases. The figure also demonstrates the preservation of a wide distribution of line-of-sight velocities within the probe volume when measuring the shear layer of the wake edge. The need for expert development of the above technique and potential difficulty in adapting to new types of signal returns is part of the motivation for development of the machine learning approach described next.




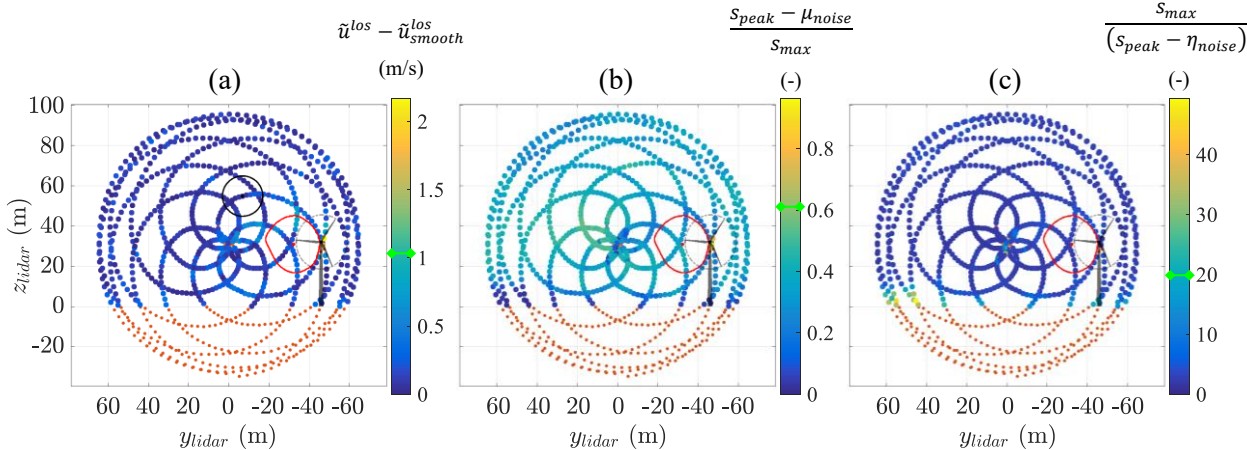

**Figure 4. Details of final filtering techniques applied between Figure 3 (b) and (c) using colormaps of the rosette scan pattern: (a) outlier detection from the difference between spectral median line-of-sight velocity, $\tilde{u}^{los}$, and smoothed spectral median line-of-sight velocity, $\tilde{u}^{los}_{smooth}$ ; (b) outlier detection from the normalized power spectral density peak prominence, $(s_{peak} - \mu_{noise})/s_{max}$ where $s_{max} = 2^{16} - 1$, and (c) filter on the reciprocal of a signal quality metric, $s_{max}/(s_{peak} - \eta_{noise})$, where $\eta_{noise}$ is the median value of the noise peaks over a region of $u^{los}$ similar as that used to calculate $\mu_{noise}$ (i.e., away from the RoI). $y_{lidar}$ and $z_{lidar}$ are the lateral and vertical coordinates, respectively, of the lidar coordinate system. Note that (a) and (b) are incorporated into a single filtering operation as described in the text while (c) constitutes a second filtering operation. The thresholds for each subfigure are indicated by the green horizontal line on the color bars.**

## 2.3. Machine Learning

The machine learning technique is an application of supervised machine learning regression via ensemble neural networks. The approach follows from the one-time construction of a high-dimensional parametric database of synthetic lidar spectra. A model of correspondence is then developed between the raw spectral shape and the QoI. The subsections below describe the

neural network architecture, the training and testing approach, and prediction confidence level.



### 2.3.1. Architecture

The network architecture is depicted in Figure 5. The individual network architecture is six hidden layers with 48 nodes each. Each node features a sigmoid symmetric

transfer function, and model learning is based on mean-square-error evaluation and backpropagation using Levenberg-Marquardt. The input layer receives the spectral magnitudes of the first 129 bins in the spectrum, which correspond to a $u^{los}$ range of 0.75-19.95 m/s and

is more than wide enough to capture the RoI of all the cases studied below.

Ensembles of the individual networks are generated to increase the regression performance by addressing the bias-variance trade off; the relatively large number of

nodes in the individual networks produces low bias estimates while cross-referencing results from multiple networks attenuates the high variance associated with such large individual networks. The ensemble training approach is a classical one of bootstrap aggregating

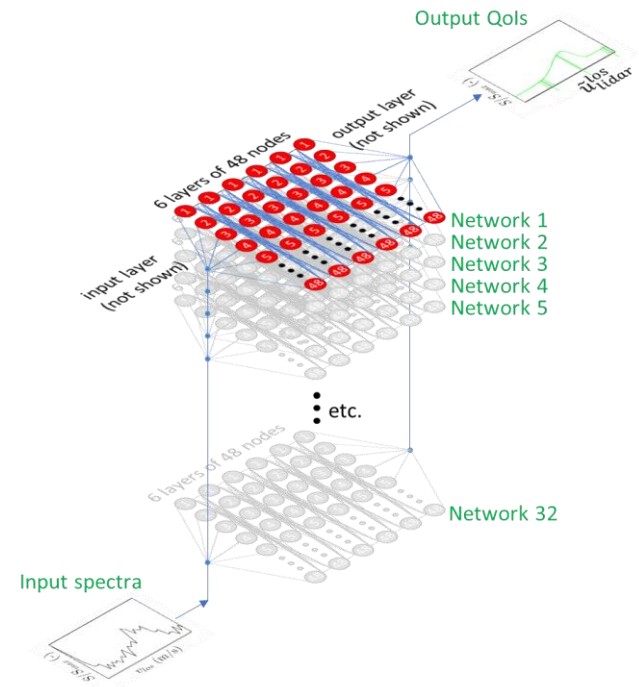

**Figure 5. Depiction of the ensemble neural network structure. Note that input and output layers are omitted for clarity.**

(often referred to as bagging) (Breiman, 1996), where $B$ individual networks are trained by bootstrapping samples with replacement from the training dataset such that the number of bootstrapped samples is the same as the training data size. The bagging approach has been found to be resistant to model misspecification and overfitting (Tibshirani, 1996). We use $B = 32$.

Once the one-time training of the $B$ individual neural networks is complete, we calculate the QoI from the median output of all $B$ individual networks in the ensemble. In our work to be shown below, the QoI is $\tilde{u}^{los}$, though it is also possible to

generate estimates for other QoIs such as the spectral standard deviation.

### 2.3.2. Training and Testing

The individual networks are implemented and trained through MATLAB's parallelized `trainNeuralNetwork` function. This function requires a training dataset, as well as a validation dataset to determine when to terminate model refinement (i.e., to determine when the model begins to lose generality and overfit the training data). A third dataset is isolated completely from

the training process to test the final model for generality. The split of training, validation, and testing data is 70, 15, and 15%, respectively.





The synthetic spectra to be used for the training, validation, and testing of the ML model are generated from full-factorial parametric sweeps through a gridded seven-

dimensional space designed to replicate the range of spectral shape parameters observed in the field. The process of replicating the range of observed shapes, which is described in Appendix A, is important since a trained model only produces valid output if the input data fall within the

distribution of the training data. Related to this aspect, several limitations of the synthetic spectra database for this effort to bear in mind are the bound on the peak prominence of the RoI, which is required to be $4\sigma_{noise}$ above $\mu_{noise}$ (see Appendix A for more on these two noise parameters), the

bound on $\tilde{u}^{los}_{lidar}$, which is set to be no less than 2 m/s, and the inclusion of only single-peaked spectra (i.e., no double-peaked spectra often found at the shear layer of a wind turbine wake). In addition to relaxing some of these constraints, future efforts might also benefit from generating

synthetic spectra that satisfy not only the range but also the probability distribution of the statistics from the field data.

The use of the ~58,000 training cases and ~13,000 validation cases produces convergence of the root-mean-square (RMS) error calculated on the ~13,000 isolated testing

cases to 0.141 m/s. Figure 6 shows the performance of the

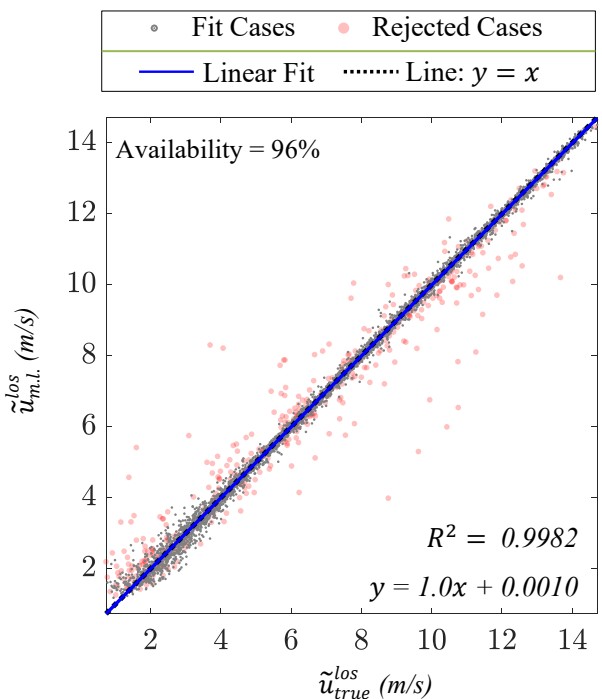

**Figure 6. Relationship between machine-learning-predicted spectral median, $\tilde{u}^{los}_{m.l.}$, and the true value, $\tilde{u}^{los}_{true}$, from the 13,383 testing cases. The red data indicate predictions with low confidence as explained in Section 2.3.3. Removing these datapoints, which was done before the regression fitting, leaves partial data availability as indicated. Note that the $y = x$ line is mostly obstructed from view by the linear fit line.**

ensemble network on the testing cases where only the datapoints in gray, which represent the predictions with highest confidence as explained in the next subsection, are used in the linear regression and RMS error calculations. The magnitude of the residuals is relatively constant with velocity except near the origin where the parameter estimation process can be complicated by the presence of the inverse function as described in Appendix A.

The variance component of error in neural networks often dominates the bias component (Geman et al., 1992), and this scenario is borne out even for our ensemble neural network, which has RMS error much larger than mean error. However, the variance error is still relatively low in the context of wind energy applications. In practice, the variance (and bias) will be shown to be larger because of the presence of inhomogeneity within the lidar probe volume.





### 2.3.3. Prediction Confidence

The ensemble strategy provides not only an estimate of a QoI via the median of the individual network outputs but also an associated estimate of the uncertainty of the QoI based on the distribution of the outputs from the individual networks. We calculate the standard error of the ensemble estimate as $\theta/(B-1)^{1/2}$, where $\theta$ is the standard deviation of the individual network outputs. This approach, which benefits from the bootstrapping performed in the training process as described above, was found to provide a better estimate of standard error from multilayer perceptrons than several other approaches reviewed

by Tibshirani (1996), and our own initial experience showed better performance with this approach than with one that trains a separate ensemble on the residual errors of the first ensemble.

In our implementation, we leverage the standard error to flag spectra that produce relatively large variation in the QoI across the ensemble members. Specifically, we set a threshold of standard error of 0.09 m/s, above which data are rejected as unreliable. This threshold provides an acceptable balance between data availability and variance error, though the tradeoff has

not yet been studied exhaustively.

### 3. Experimental Techniques

### 3.1. Facility

A validation case for the lidar processing techniques is derived from data at the Scaled Wind Farm Technology (SWiFT) facility in Lubbock, Texas, USA as illustrated in Figure 7. The site features level terrain with minimal surface roughness, and

characterization of the atmospheric conditions is given in Kelley and Ennis (2016) with recent benchmarking and validation activities given in Doubrawa et al. (2020) and Hsieh (2021).





Each of the three V27 wind turbine rotors on the site are 27 m in diameter, $D$, and have hub heights of 31.5 m. Two meteorological towers are positioned 2.5 $D$ ahead of the frontline turbines relative to the prevailing wind direction as shown in Figure 7. Data were taken with the lidar scanning over the meteorological towers both with the rotors stationary and with the WTGa1 rotor operating, and we hereafter refer to these cases as the inflow and waked cases, respectively. These data derive from a 2016-2017 test campaign, most of the data for which has been released into the public domain through the A2e Data Archive and Portal (2019).

**3.2. Ultrasonic Anemometers**

SATI Series 'A' style probe ultrasonic anemometers from ATI Technologies, Inc. are located at 10.1 m, 18.3 m, 31.9 m, 45.4 m, and 58.3 m above the ground on the METa1 and METb1 meteorological towers. The anemometers sample data at

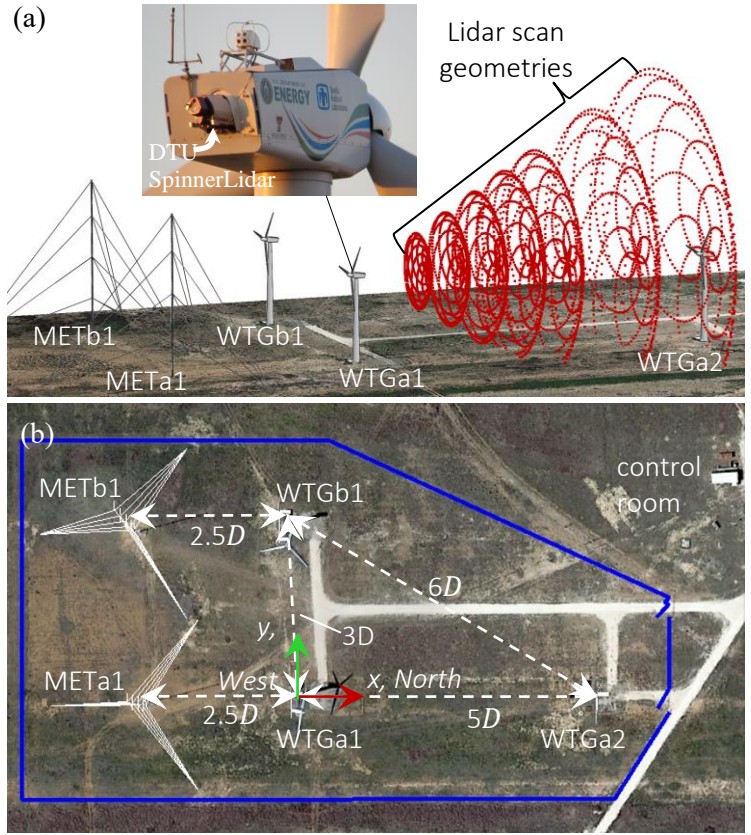

Figure 7. (a) Rendering of the SWiFT facility in Lubbock, Texas, USA. The nacelle of WTGa1 was outfitted with a rear-mounted DTU SpinnerLidar, which scanned the flow at different focus lengths according to the rosette patterns shown in red. Rendering from Doubrawa et al. (2020). (b) Planform view of the site where $D$ = 27 m. Adapted from Herges and Keyantuo (2019).

100 Hz and measure $u_{sonic}$, $v_{sonic}$, and $w_{sonic}$, which are the velocity components in the site reference frame according to the coordinate system of Figure 7. The manufacturer-quoted accuracy of the $u_{sonic}$ and $v_{sonic}$ components is $\pm 0.01$ m/s. The total uncertainty of the horizontal wind direction measurement derived from the sonic anemometers is estimated at 1.22°.

Occasional spurious spikes in the signal are removed in pre-processing using a median absolute deviation filter with a length of 10,000 data points, or 100 seconds. Data lying more than five standard deviations away from the median are also removed. In addition to these standard quality control processing techniques used at SWiFT, this effort uses shape-preserving piecewise cubic interpolation across any spans of data up to 1 s in length that have been omitted either due to the removal process noted above or to malfunction of the instrumentation. Longer segments of instrument cutout were removed from





consideration to be used in this study. No blockage correction was made for the presence of the tower or anemometers in this study.

### 370   3.3.  Laser Anemometer (Lidar)

Scans from the DTU SpinnerLidar (Mikkelsen et al., 2013) mounted on WTGa1 will be considered below. The beam pointing accuracy of the instrument is not quantified exactly, but the pointing direction has been verified with infrared photogrammetry in the lab (Herges et al., 2017), and the beam position in the field is known from the combination of Theodolite total station measurements of the lidar location in the stationary nacelle, the lidar accelerometers, and the turbine yaw heading. The scans
of interest were focused $2.5D$ from WTGa1. At this focus length, the full-width half-maximum (FWMH) averaging length of the beam is 8.45 m as defined by a truncated Gaussian weighting function. Integrating the weighting function over $16 \times$ FWHM corresponds to over 99% of the full weight along the beam path; see Debnath et al. (2019) for more information. The probe volume averaging acts as a low-pass filter for the timeseries of $\tilde{u}^{los}$ from the lidar, but the filtered small-scale turbulence content is returned in each scan as additional power density spectral width, which in some cases can be used to improve
turbulence estimates (Branlard et al., 2013; Peña et al., 2017).

The rosette scan patterns of the SpinnerLidar are completed in 2-4 s and consist of 984-1968 measurement locations, some of which are eliminated from the measurement domain when the focus distance falls below the surface of the ground. Within each scan, the lidar samples at 100 MHz, and power spectra are calculated from sequences of 512 samples to yield 256 fast-Fourier-transformed bins so that the returned power spectrum for each measurement location is the average of ~400
consecutive spectra.

The $CNR$ is calculated from the lidar spectra according to a wideband defined as Eq. (2):

$$CNR = \left( \mu_{noise} \left( u_{max}^{los} - u_{min}^{los} \right) \right)^{-1} \int_{u_{min}^{los}}^{u_{max}^{los}} (s - \mu_{noise}) \; du^{los} \tag{2}$$

where $u_{\blacksquare}^{los}$ is the minimum or maximum line-of-sight velocity sensed by the lidar according to the subscript. For the lidar used here, $u_{max}^{los}$ is 38.40 m/s and $u_{min}^{los}$ is 0.75 m/s (velocities lower than 0.75 m/s are removed due to high relative intensity noise (Lindelöw, 2007)). Practically, the integral in Eq. (2) is evaluated discretely using trapezoidal integration over the bin width
of 0.15 m/s. $CNR$ calculations are performed using the advanced filtering technique output spectra since this provides a more complete RoI than the thresholded spectra and since the machine learning does not produce output spectra to integrate.

### 3.4.  Pre-Processing

Our work compares estimated velocities from the lidar spectra to point measurements from the sonic anemometers for cases when the lidar beam passed within a certain distance of the anemometers. Several steps are necessary to enable an appropriate
and meaningful comparison as described below.





### 3.4.1. Bin Selection

Similar to Gottschall et al. (2012), filtering of the 10-minute bins for the present campaign was performed to isolate cases of specific interest. Several filters were applied to all 10-minute-averaged bins. Bins without the lidar activated were disregarded, as were bins with the yaw heading more than 60° from zero since the lidar measurement volume would not overlap the

meteorological tower for those cases. The wind direction was also constrained to be within a certain tolerance of the line of sight of the lidar beam so that the lidar could resolve a significant component of the wind speed, and this tolerance was 30° and 60° for the inflow and waked cases, respectively. For the inflow cases, which have a larger database than the waked cases, additional filters were applied requiring all five sonic anemometers to be functioning and limiting the maximum wind speed to 6 m/s. This second constraint is imposed because the lower velocity cases are the ones for which high-fidelity processing

becomes most difficult in the presence of solid interference, which is a major focus of this article. Both inflow and waked cases, however, were required to have maximum wind speed greater than 2 m/s.

The filtering resulted in 69 10-minute bins for the inflow cases as shown in Table 1 and six 10-minute bins for the waked cases as shown in Table 2. These data cover multiple seasons, spanning from January 16 to July 11, 2017. Data cover a range of stability states of the atmospheric boundary layer (ABL) as can be inferred from the more than order of magnitude variation

in standard deviation of wind speed, which corresponds to turbulence intensities between 1-29% for Table 1 and 6-21% for Table 2.

**Table 1**. Summary statistics of the 69 10-minute bins used for the validation study of inflow data. Data shown correspond to the sonic anemometer at 31.9 m. The values $ws$ and $wd$ are the horizontal wind speed and wind direction, respectively. The value $\gamma$ is the turbine yaw heading, which corresponds to clockwise rotation in the reference frame of Figure 7.

|  | $\overline{u_{sonic}}$ | $\overline{v_{sonic}}$ | $\overline{w_{sonic}}$ | $\overline{ws_{sonic}}$ | $\sqrt{\overline{(ws_{sonic} - \overline{ws_{sonic}})^2}}$ | $\overline{wd_{sonic}}$ | $\sqrt{\overline{(wd_{sonic} - \overline{wd_{sonic}})^2}}$ | $\bar{\gamma}$ |
|---|---|---|---|---|---|---|---|---|
|  | (m/s) | (m/s) | (m/s) | (m/s) | (m/s) | (°) | (°) | (°) |
| Minimum | 2.69 | -1.31 | -0.26 | 2.82 | 0.08 | 165.01 | 0.94 | 346.98 |
| Mean | 4.56 | 0.18 | 0.02 | 4.63 | 0.42 | 177.81 | 5.25 | 347.06 |
| Maximum | 5.96 | 1.28 | 0.34 | 5.98 | 1.37 | 194.45 | 21.73 | 347.11 |

**Table 2**. Summary statistics of the six 10-minute bins used for the validation study of waked data. Data shown correspond to the sonic anemometer at 31.9 m. The values $ws$ and $wd$ are the horizontal wind speed and wind direction, respectively. The value $\gamma$ is the turbine yaw heading, which corresponds to clockwise rotation in the reference frame of Figure 7.





| | $\overline{u_{sonic}}$ | $\overline{v_{sonic}}$ | $\overline{w_{sonic}}$ | $\overline{ws_{sonic}}$ | $\sqrt{\overline{(ws_{sonic} - \overline{ws_{sonic}})^2}}$ | $\overline{wd_{sonic}}$ | $\sqrt{\overline{(wd_{sonic} - \overline{wd_{sonic}})^2}}$ | $\bar{\gamma}$ |
|---|---|---|---|---|---|---|---|---|
| | (m/s) | (m/s) | (m/s) | (m/s) | (m/s) | (°) | (°) | (°) |
| Minimum | -6.29 | -3.27 | -0.20 | 4.24 | 0.25 | 26.83 | 3.59 | 324.95 |
| Mean | -5.37 | -1.07 | -0.04 | 5.89 | 0.67 | 274.81 | 6.51 | 338.10 |
| Maximum | -4.02 | 2.96 | 0.11 | 6.84 | 1.07 | 342.46 | 9.41 | 18.55 |

### 3.4.2. Within-Bin Filtering

Within each bin scans were removed when the lidar was focused at distances other than $2.5D$ (i.e., the nominal distance to the meteorological tower), when the sonic anemometers were malfunctioning, and when there was more than 2 m of separation between the lidar beam path and the sonic anemometer in the $x = -2.5D$ plane as described in the following subsection. The within-bin filtering resulted in net comparison times of at least 355 and 14 minutes per sonic anemometer location for the validation study of the inflow and waked cases, respectively (the small sample size of the waked cases will be discussed below).

Various levels of other filtering, or sub-binning, are also explored in Section 4 to bin data on certain QoIs. The highest level of such sub-binning determines whether an individual return includes solid interference or not. This determination is performed similarly to the thresholding technique in Section 2.1; a return is designated as having solid interference if the first useable velocity bin of the spectrum lies above the threshold described by Eq. (1), which often occurs when solid interference is present. Other sub-binning operations include those based on the lidar $CNR$ as well as on the time-local standard deviation of velocity from the sonic anemometers. For the latter, calculations are made using a running standard deviation with a window span corresponding to $16 \times$ FWHM (the relationship between time window span and probe length is approximated by invoking Taylor's hypothesis to translate the time stamps of each sonic anemometer reading to horizontal distances from the lidar focus point for any given lidar scan). Note that the resulting quantity is related to the streamwise turbulence intensity by division of the streamwise velocity, but the absolute magnitude of the fluctuations in the atmosphere are considered here to be more relevant than the conventional normalized quantity in the context of the comparisons to be made below.

### 3.4.3. Spatio-Temporal Syncing

Once a 2-4 s scan window has been deemed valid for the validation analysis, a process is used to isolate the exact scan indices within each window when the lidar beam was pointed closest to each of the sonic anemometers. The lidar beam position is known in the coordinate system depicted in Figure 7 as described in Section 3.3. For the waked cases, the turbine yaw setting was variable as indicated in Table 2, which resulted in high variability of the closest-passing scan indices within the rosette

scan pattern. For the inflow cases, the turbine setting was usually fixed at 347°, so the closest-passing scan index was more predictable. For all cases, the closest-passing scan index was only retained for this work if there was less than 2 m of separation

between the lidar beam path and the sonic anemometer in the $x = -2.5D$ plane as shown in Figure 8. Note that for scan indices with $\delta \neq 0$, the focus point of the probe volume is slightly offset from the $x = -2.5D$ plane according to the $2.5D$ radius of the

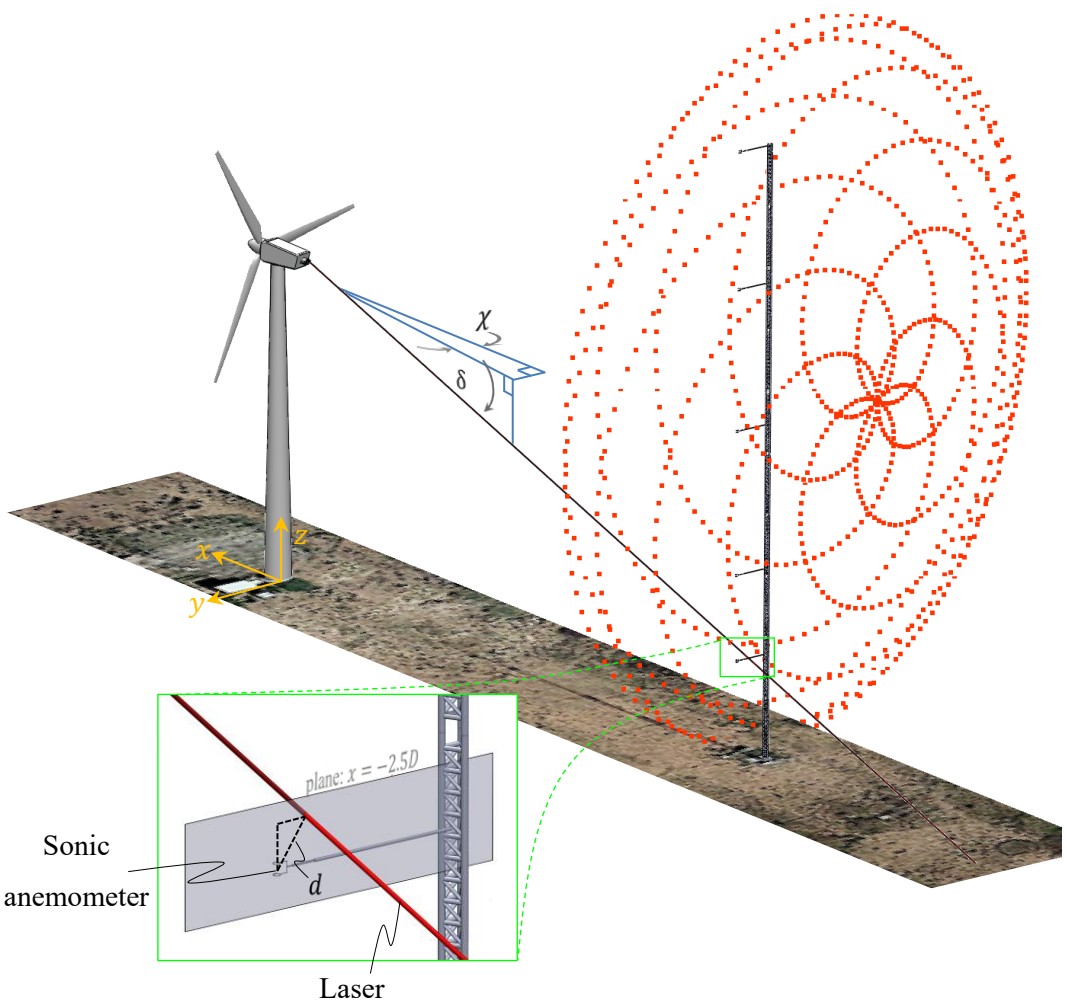

**Figure 8. Rendering of WTGa1 and META1, where the former is yawed to 347° (i.e., nearly 180° from the orientation shown in Figure 7) to cover the latter with the lidar scan pattern. Guy wires have been removed for clarity. The value $\chi$ is the horizontal directional offset of the lidar beam from the $-x$ axis, and $\delta$ is the elevation angle of the beam, which becomes –18.2°, -11.6°, 0.2°, 11.7°, and 22.0° when the beam is pointed at the center of the five respective anemometers at $z = 10.1$ m, 18.3 m, 31.9 m, 45.4 m, and 58.3 m. The maximum distance, $d$, in the $x = -2.5D$ plane between the lidar beam and the center of the sonic anemometer probe for the considered data is 2 m. Note that the lidar beam is thickened for clarity; the actual beam diameter at the waist is 2.7 mm for the $2.5D$ focus point.**



hemispherical scan geometry. Once the closest-passing index is identified, the data from the nearby sonic anemometer is interpolated cubically in time to the moment when the lidar beam sampled near it.

### *3.4.4.* **Projection of Velocity Components**

Although the lidar and sonic anemometer feature significantly different measurement volumes and thus can never be compared to the highest degree of confidence, projecting the sonic anemometer velocity data onto the line of sight of the lidar beam is an important step to removing some of the uncertainty of the comparison. Projection was performed for each of the five closest individual scan indices identified in Section 3.4.3. 3.4.2. to produce the line-of-sight velocity, $u_{sonic}^{los}$, for each sonic anemometer according to Eq. (3):

$$u_{sonic}^{los} = [\ \cos(\chi)\cos(\delta) \quad \sin(\chi)\cos(\delta) \quad \sin(\delta)] \begin{pmatrix} u_{sonic} \\ v_{sonic} \\ w_{sonic} \end{pmatrix}, \tag{3}$$

where $\chi$ and $\delta$ are the angles defined by Figure 8.

### **3.5. Time-Averaging Error**

The uncertainty bands on ensemble data shown at various instances in Section 4 below correspond to the statistical time-averaging error (i.e., random uncertainty) and are derived from 10,000 bootstrap resamples with a 95% confidence level (Benedict and Gould, 1996).

### **4. Validation Results**

This section presents the results of the validation study. Section 4.1. gives examples to demonstrate the processing techniques qualitatively, while Sections 4.2. and 4.3. give the validation exercises for the inflow and waked cases, respectively. Due to the larger sample size for the inflow cases, we spend significantly more time analysing these cases.

### **4.1. Processing Examples**

An example of an instantaneous comparison between a lidar return and sonic anemometer data is shown in Figure 9. This example case illustrates several features observed throughout the full dataset.

    First, the turbine is yawed at 347°, as it was for all of the inflow cases, and the location of the five closest lidar scan indices in Figure 9(a) relative to the five sonic anemometers is thus representative of most of the inflow cases, the only exception being several 10-minute bins that were measured with the lidar mounted at a non-zero yaw of –15.1°, which caused the closest

lidar scan indices to fall at the plane of symmetry of the rosette scan pattern. For the handful of waked cases, the turbine yaw setting was continuously variable, which led to a wider range of scan indices being used for the validation.



**Figure 9.** Comparison of lidar processing techniques for a single 2 s inflow scan at 2:38:10 (local time) on February 9, 2017. In (a), the instantaneous location of the lidar scan pattern (in black) is overlaid on the met tower (in gray) with the scan location closest each respective sonic anemometer highlighted (in red). In (b), the sonic line-of-sight velocity, $u_{sonic}^{los}$, and lidar spectral median line-of-sight velocity, $\tilde{u}_{lidar}^{los}$, are plotted verses height, where the sonic anemometer data has been temporally interpolated to the same instant as the lidar passing. The uncertainty bands indicate the sonic anemometer instrument's quoted uncertainty. Subfigures (c)-(g) show the individual scaled lidar spectra, $s/s_{max}$ where $s_{max} = 2^{16} - 1$, for each of the index locations in (a) and (b) where solid lines (−) indicate spectral magnitude and vertical dashed lines (--) indicate the line-of-sight velocity estimates. Subfigures (c)-(g) also show the raw lidar spectra, which have no corresponding entries in (b) since the raw data precedes the MFE process.





In Figure 9(b), the velocity estimates of the sonic anemometer indicate a roughly logarithmic boundary layer profile at this instant, and the disagreements between $u_{sonic}^{los}$ and $\tilde{u}_{lidar}^{los}$ are congruous with our understanding of the lidar measurement

principles and processing. The smallest error between the two instruments occurs at the middle lidar location corresponding to $\delta = 0.2°$ while there is added error away from this $\delta$ setting, both because the lidar probe volume samples through a vertically nonhomogeneous ABL as well as because of truncation of the spectra at low velocities by the unusable bins at the beginning of the spectra.

Insight to the comparison of the three lidar processing techniques in Figure 9(b) is provided with the help of subfigures (c)-

(g), which show the spectral returns for each index. In general, we find that the advanced filtering and machine learning techniques have similar estimates of $\tilde{u}_{lidar}^{los}$, while the thresholding technique shows significant deviations for indices 532 (Figure 9(c)) and 222 (Figure 9(e)). As before in Figure 9(b), the thresholding technique gives no estimate whatsoever for index 532 in Figure 9(c) since the high magnitude of the first useable velocity bin flags this spectra as a full solid return, which the thresholding technique therefore rejects as described in Section 2.1. This solid return is due to ground interference, which

is a common scenario for the scan indices with relatively large negative $\delta$ like index 532, depending on the scattering behavior of the laser at the exact location of intersection with the ground. For index 222 in Figure 9(e), the thresholding technique does give an estimate, but the estimate is strongly biased because of a partial solid return. This return is due to interference from the meteorological tower or sonic anemometer itself and is a common occurrence in our validation dataset because of the proximity of the lidar scan to the meteorological tower, particularly for the $\delta = 0.2°$ indices. Both the advanced filtering and machine

learning techniques successfully ignore the signature of this solid interference and estimate $\tilde{u}_{lidar}^{los}$ near to $u_{sonic}^{los}$.

It is also worth noting the small differences in spectral shape between the thresholding and advanced filtering techniques above the threshold limit, which are due to the bilateral smoothing process across adjacent scans of the advanced filtering technique. The machine learning technique implicitly performs its own smoothing operation (without regard to adjacent scans), but no visualization of this smoothing is possible since the machine learning technique generates no output spectra.

Next, we show sample processed data from 10-minute bins in Figure 10 that illustrate several points about the time series of processed lidar data. Figure 10(a) represents a bin with low turbulence intensity and one for which $\tilde{u}_{lidar}^{los}$ from the three lidar processing techniques tracks $u_{sonic}^{los}$ qualitatively well. Figure 10(b) shows a case of higher turbulence intensity, where all three processing methods again perform similarly well, though small discrepancies between methods are observable. Figure 10(c) shows a bin where solid returns produce a number of instants where none of the three processing techniques yield an

estimate. Figure 10(d) is a wake case, and there are a handful of instances where the thresholding and machine learning approaches again do not produce estimates because of strong solid returns from the meteorological tower. While the advanced filtering technique produces higher data availability for this bin, a stronger bias is detected in these results for the estimates between 03:14 and 03:16 than for the other two techniques. Note that the first half of the bin is removed from the comparison because the separation between the lidar beam and sonic anemometer in the $x = -2.5D$ plane exceeds the 2 m tolerance.





Other than the cases with solid returns, the agreement of all three processing methods with the reconstructed data is qualitatively good. The following sections provide a more quantitative statistical perspective of the performance of each of the processing methods.

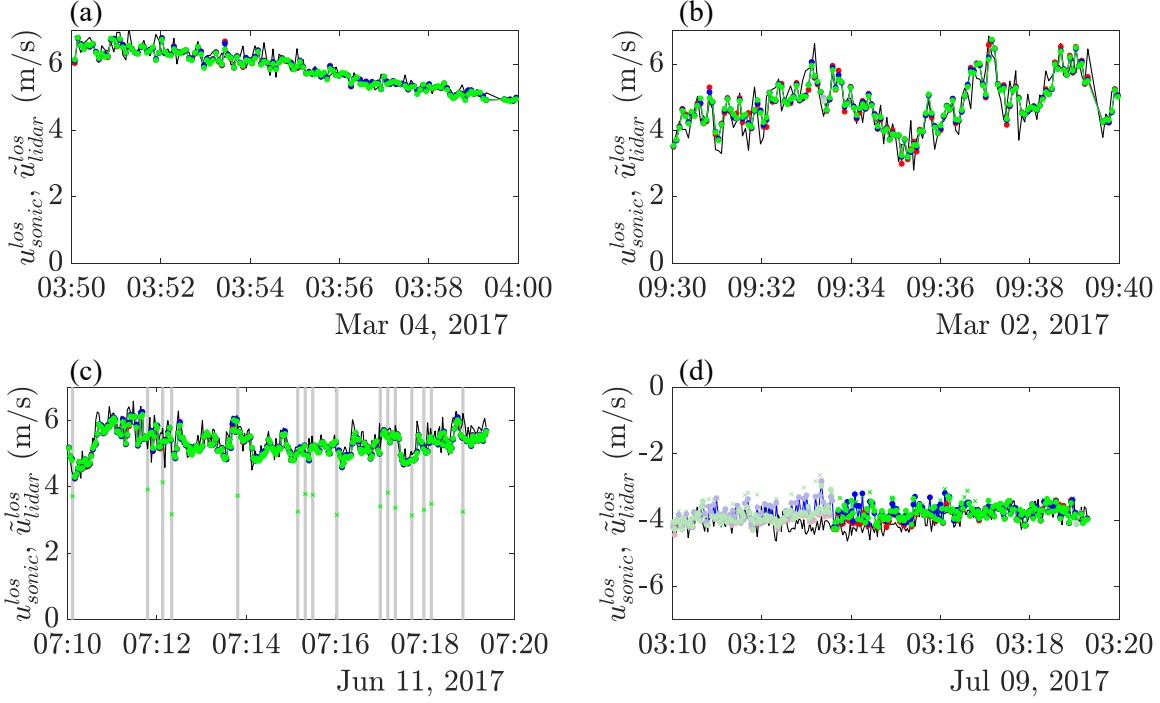

**Figure 10.** Time series comparisons of spectral median lidar line-of-sight velocity, $\tilde{u}_{lidar}^{los}$, with the nearby sonic anemometer line-of-sight velocity, $u_{sonic}^{los}$, for four sample 10-minute bins at scan index positions with $\delta = 0.2°$. Subfigures (a)-(c) are inflow cases, and (d) is a wake case. Faded color indicates scans where the distance between the lidar beam and the center of the sonic anemometer in the $x = -2.5D$ plane was larger than the 2 m tolerance. In the machine learning results, a green "x" indicates a removed reading due to low confidence as explained in Section 2.3.3. A vertical gray line indicates an instant when none of the three of the processing techniques returned a reading.

—— Sonic anemometer
—— Lidar (thresholding)
—— Lidar (adv. filtering)
—— Lidar (machine learning)





 **4.2. Inflow Cases**

This section contains the results from our analysis of the 69 bins with inflow cases described in Table 1. First, we offer insight on the trends in the lidar errors for the cases without and with solid interference, which have total return counts of 47,927 and 7,183, respectively. Next is a description of the practical significance of these trends for wind turbine applications.

*4.2.1.* **Error Trends without Solid Interference**

 Considered first is the inflow data filtered to exclude any returns with solid interference present as described in Section 3.4.2. Figure 11 shows scatterplots of all such results differentiated by height for the three processing techniques. The similarity of the three subfigures is expected, and all three techniques produce roughly the same mean and random error when no solid interference is present. Notably, the data availability for the machine learning technique is 3% lower than for the other two techniques, though this gap might be helped with an improved machine learning architecture and training scheme. While the  overall performance between the three techniques for these cases of non-interfered returns appears fairly similar, further analysis is warranted to better understand several nuances of the QA/QC problem.

The sources of bias observed in Figure 11 are several, though only one is likely related to the processing technique. This processing-related bias source is the truncation of RoIs that fall at least partially over the unusable velocity bins at the beginning of the spectra. This truncation will artificially increase $\tilde{u}_{lidar}^{los}$ for low velocities, which is indeed the trend observed comparing

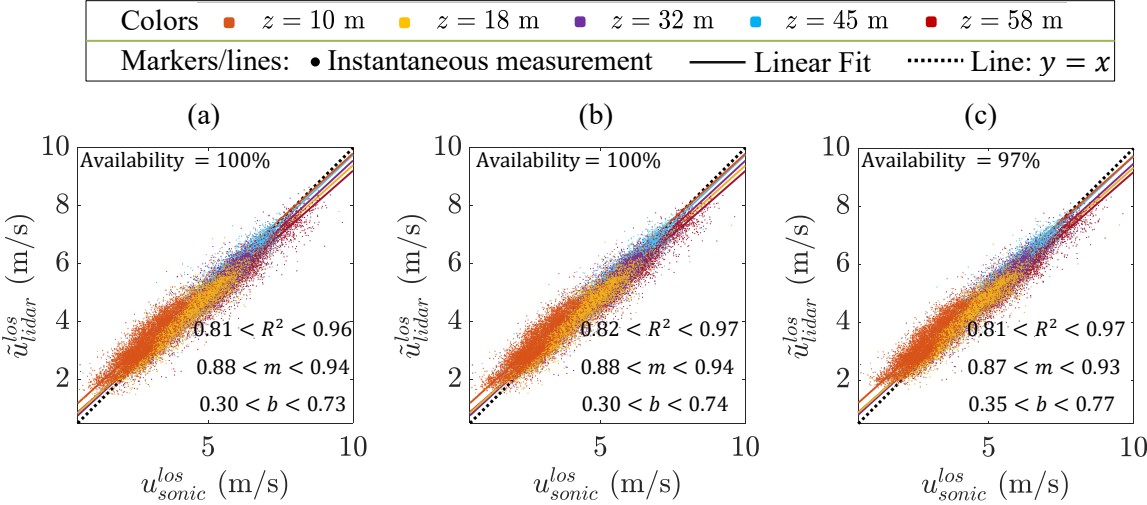

**Figure 11.** Instantaneous lidar line-of-sight spectral median velocity estimates, $\tilde{u}_{lidar}^{los}$, versus sonic anemometer line-of-sight velocity estimates, $u_{sonic}^{los}$, for inflow cases without solid interference from the (a) thresholding, (b) advanced filtering, and (c) machine learning techniques. The variation shown for the coefficient of determination value, $R^2$, the linear fit slope, $m$, and the linear fit offset, $b$, correspond to the ranges observed across the fits at all five comparison heights.





$\tilde{u}_{lidar}^{los}$ and $u_{sonic}^{los}$ at the 10 m height (i.e., where velocities are the lowest on average). While the machine learning technique offers the possibility to eliminate such a bias as noted in Appendix A, we do not observe any practically significant differences in the mean offset or slope of the linear trend lines between processing techniques in the dataset in Figure 11.

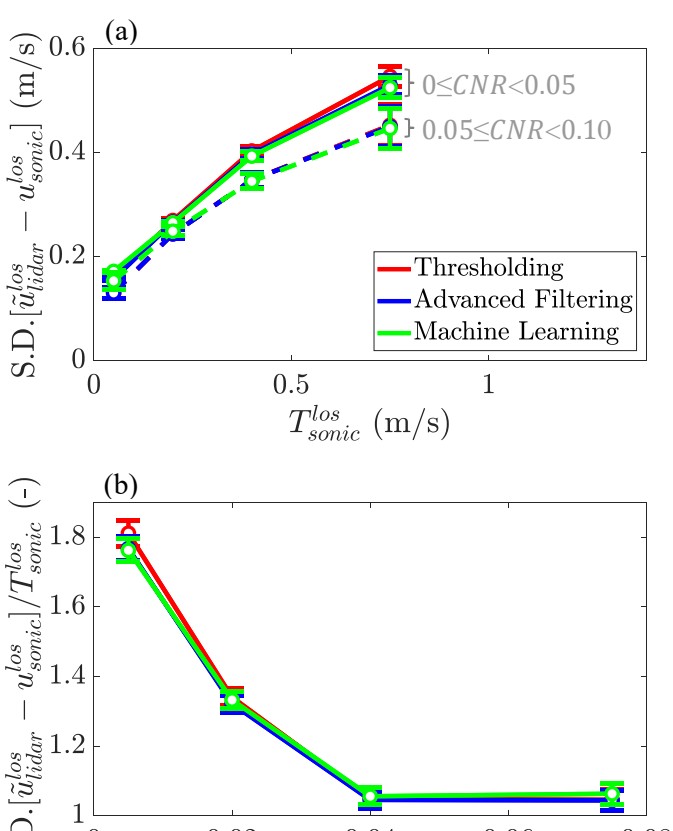

Figure 12. Convergence of the standard deviation, S.D., of error with (a) magnitude of time-local turbulent fluctuations, $T_{sonic}^{los}$, and (b) carrier-to-noise ratio, $CNR$, for inflow cases without solid interference across the four higher comparison heights. $T_{sonic}^{los}$ is the running standard deviation of velocity from the sonic anemometer along the line-of-sight based on the filtering window described in Section 3.4.2. The $CNR$ values are calculated from the advanced filtering technique's output spectra as described in Section 3.3. Data plotted derive from all the comparison heights except the bottom, $z = 10.1$ m position.

There remains a bias of $\tilde{u}_{lidar}^{los}$ even at higher velocities in Figure 11, which suggests the presence of another, most likely more dominant source of bias. This source stems from the difference between the spatial distribution of the measurement volumes of the sonic anemometer and the lidar. For example, the asymptotic character of the ABL mean velocity profile with increasing height can produce a reduction in the height-averaged $\tilde{u}_{lidar}^{los}$ compared to a point at the center of the probe volume (e.g., a sonic anemometer at the height of the lidar focus point), which is in fact the trend observed for the comparisons with the sonic anemometer at 58 m (see also Figure 9(b)).

Another source of bias is that introduced due to local flow blockage around the sonic anemometers, boom arms, and meteorological tower, which may not be sensed by the lidar beam depending on its relative position. Furthermore, there is clearly potential for internal bias in the sonic anemometers and lidar (Lindelöw-Marsden, 2009).

Related to scatter, two main sources in Figure 11 are the spectral width resulting from turbulence within the lidar probe volume as well as amplitude noise. Based on the findings of Simley et al. (2014), we expect the former to be larger than the latter, especially in high turbulence conditions since the RMS error due to the lidar's probe

550 volume averaging scales linearly with turbulence magnitude. An approximately linear scaling is validated in Figure 12(a), which bins the data for four of the five comparison heights based on the running standard deviation of velocity from the sonic anemometer along the line-of-sight, $T_{sonic}^{los}$, as





described in Section 3.4.2 (the bottom comparison height was omitted because of irregular trends possibly related to proximity of the RoIs to the unusable bins of the lidar). Figure 12(a) can be interpreted to suggest that much of the random error is a function of turbulence magnitude. On the other hand, one can extrapolate the trend to $T_{sonic}^{los} = 0$ to estimate that there exists a baseline standard deviation of error of roughly 0.08 to 0.15 m/s, which is primarily due to the interference of amplitude noise on the parameter estimation problem. Because of the coarse resolution of the binning and the possibility of the trendline flattening out as $T_{sonic}^{los} \to 0$ due to the discretization of the lidar spectra, we conservatively take the value at the first $T_{sonic}^{los}$ bin to be the estimated contribution of amplitude noise, which is 0.13 and 0.16 m/s for the higher and lower ranges of $CNR$, respectively, of the thresholding and advanced filtering techniques. These values are 0.15 and 0.17 m/s, respectively, for the machine learning technique. Note that the uncertainty of the sonic anemometer velocity, quoted at $\pm 0.01$ m/s for $u_{sonic}$ and $v_{sonic}$, is small relative to the above values.

The effect of the amplitude noise on random error, which is a function in part of the processing technique, can be drawn out explicitly as in Figure 12(b), where the influence of $T_{sonic}^{los}$ is nominally removed by the scaling of the ordinate and where the standard deviation of errors has been binned on $CNR$ instead. Figure 12(b) shows a principle derived from the Cramer-Rao lower bound (Rye and Hardesty, 1993), which is that the minimum attainable variance of the spectral estimation process is an inverse function of $CNR$. The slightly better parameter estimation performance of the advanced filtering versus the thresholding technique is a result of (1) the larger effective $CNR$ of the RoI and (2) the bilateral filtering of the advanced filtering approach. The increase in error for the lower two $CNR$ levels of the thresholding technique compared to the advanced filtering technique is expected since the thresholding process removes a larger and larger percentage of the signal as $CNR \to 0$. The estimation performance of the machine learning technique is better than that of the other two techniques at low $CNR$ and worse than that of the other two techniques at high $CNR$, though the variations in performance between techniques are relatively small for cases without solid interference.

The overall performance of each lidar processing technique as a function of height is tabulated in Table 3. Except for the 10 m position, the bias errors are smaller than the random errors and are consistent between the three processing techniques, which is expected based on the results of Figure 11. The standard deviations of the errors are between 0.23 and 0.33 m/s for the advanced filtering and machine learning techniques while the thresholding technique has 1-3% higher values. Between the advanced filtering and machine learning techniques, the former is overall the more effective within the bounds of the data considered in this study because of higher data availability and slightly better noise rejection.





**Table 3.** Performance of lidar processing techniques versus sonic anemometer for inflow cases without solid interference. The processing abbreviations are threshold ($th.$), advanced filter ($a.f.$), and machine learning ($m.l.$). S.D. refers to standard deviation.

| Height (m) | | Mean [ $\tilde{u}_{lidar}^{los} - u_{sonic}^{los}$ ] (m/s) | | | S.D. [ $\tilde{u}_{lidar}^{los} - u_{sonic}^{los}$ ] (m/s) | | | Availability (-) | | |
|---|---|---|---|---|---|---|---|---|---|---|
| | | *th.* | *a.f.* | *m.l.* | *th.* | *a.f.* | *m.l.* | *th.* | *a.f.* | *m.l.* |
| 10 | | 0.47 ±0.01 | 0.48 ±0.01 | 0.48 ±0.01 | 0.34 ±0.01 | 0.33 ±0.01 | 0.33 ±0.01 | 100% | 100% | 97% |
| 18 | | 0.00 ±0.01 | 0.00 ±0.01 | 0.00 ±0.01 | 0.29 ±0.01 | 0.28 ±0.01 | 0.28 ±0.01 | 100% | 100% | 97% |
| 32 | | -0.04 ±0.00 | -0.04 ±0.00 | -0.05 ±0.01 | 0.25 ±0.01 | 0.24 ±0.01 | 0.25 ±0.01 | 100% | 100% | 97% |
| 45 | | 0.07 ±0.00 | 0.07 ±0.00 | 0.07 ±0.00 | 0.24 ±0.01 | 0.23 ±0.01 | 0.23 ±0.01 | 100% | 100% | 96% |
| 58 | | -0.19 ±0.01 | -0.19 ±0.01 | -0.20 ±0.01 | 0.33 ±0.01 | 0.32 ±0.01 | 0.32 ±0.01 | 100% | 100% | 98% |
| Combined | | 0.03 ±0.00 | 0.03 ±0.00 | 0.02 ±0.00 | 0.35 ±0.00 | 0.35 ±0.00 | 0.35 ±0.00 | 100% | 100% | 97% |

### *4.2.2.* **Error Trends with Solid Interference**

Figure 13 shows the data and linear fit of the comparison of all inflow cases with the solid interference flag. The lower coefficient of determination values, $R^2$, of the thresholding technique are primarily a consequence of a handful of partial solid returns that are not filtered out and that manifest as the outliers near the bottom of Figure 13(a). As described for the data without solid interference, a bias again exists for all three processing techniques near lower velocities.

The tabulated data in Table 4 shows several of the same trends as described for the non-interfered data in Table 3. Namely, the bias errors are generally much smaller than the random errors, and the advanced filtering and machine learning techniques outperform the thresholding technique in terms of random error. The notable differences from Table 3 are that the standard deviation of the errors for the advanced filtering and machine learning techniques have a larger range from 0.08 to 0.38 m/s (as compared to 0.23 to 0.33 m/s before), and that the thresholding technique now has up to 154% higher values (as compared to a maximum of 3% before) due mostly to the handful of outliers described for Figure 13(a). The thresholding technique thus

has much poorer performance than the other two techniques in terms of random error, as well as in terms of data availability, which is around 30% lower than for the other techniques over all comparison locations. Between the advanced filtering and machine learning techniques, the machine learning provides an average improvement of 0.02 and 0.01 m/s for the mean and standard deviation of error, respectively, across all comparison heights, but these small gains are traded for 6% lower data

availability across all comparison heights.





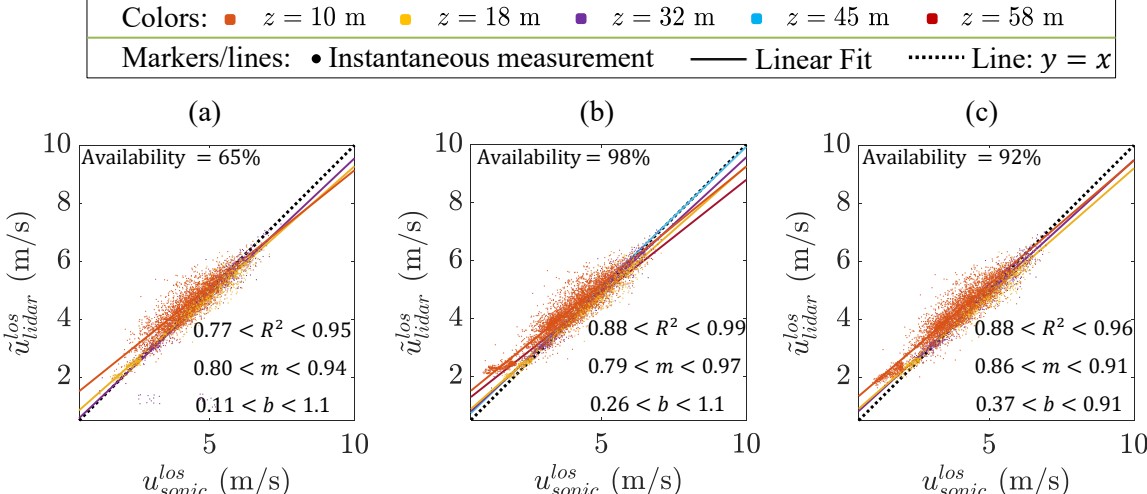

**Figure 13. Instantaneous lidar line-of-sight spectral median velocity estimates, $\tilde{u}_{lidar}^{los}$, versus sonic anemometer line-of-sight velocity estimates, $u_{sonic}^{los}$, for inflow cases with solid interference from the (a) thresholding, (b) advanced filtering, and (c) machine learning techniques. The variation shown for the coefficient of determination value, $R^2$, the linear fit slope, $m$, and the linear fit offset, $b$, correspond to the ranges observed across the fits at all five comparison heights.**

**Table 4.** Performance of lidar processing techniques versus sonic anemometer for inflow cases with solid interference. The processing abbreviations are threshold ($th.$), advanced filter ($a.f.$), and machine learning ($m.l.$). S.D. refers to standard deviation.

| Height (m) | | Mean [ $\tilde{u}_{lidar}^{los} - u_{sonic}^{los}$ ] (m/s) | | | S.D. [ $\tilde{u}_{lidar}^{los} - u_{sonic}^{los}$ ] (m/s) | | | Availability (-) | | |
|---|---|---|---|---|---|---|---|---|---|---|
| | | *th.* | *a.f.* | *m.l.* | *th.* | *a.f.* | *m.l.* | *th.* | *a.f.* | *m.l.* |
| 10 | | 0.29 ±0.02 | 0.41 ±0.01 | 0.37 ±0.01 | 0.39 ±0.01 | 0.38 ±0.01 | 0.36 ±0.01 | 56% | 98% | 96% |
| 18 | | -0.10 ±0.01 | -0.08 ±0.01 | -0.09 ±0.01 | 0.28 ±0.01 | 0.27 ±0.01 | 0.28 ±0.01 | 93% | 98% | 89% |
| 32 | | -0.16 ±0.04 | -0.02 ±0.01 | -0.04 ±0.01 | 0.57 ±0.10 | 0.22 ±0.01 | 0.24 ±0.01 | 63% | 97% | 83% |
| 45 | | N/A | 0.10 ±0.06 | N/A | N/A | 0.08 ±0.04 | N/A | 0% | 55% | 0% |
| 58 | | N/A | -0.13 ±0.29 | N/A | N/A | 0.29 ±0.24 | N/A | 0% | 60% | 0% |
| Combined | | 0.07 ±0.01 | 0.21 ±0.01 | 0.19 ±0.01 | 0.46 ±0.03 | 0.40 ±0.01 | 0.39 ±0.01 | 65% | 98% | 92% |

Figure 14 gives example spectra from the cases with solid interference to illustrate several features and deficiencies of the different processing techniques. Figure 14(a) is a common case, similar to the one shown in Figure 9(c), where the thresholding





technique does not make a prediction because of overwhelming solid interference, but the advanced filtering and machine

learning techniques successfully produce a value of $\tilde{u}_{lidar}^{los}$ approximately equal to $u_{sonic}^{los}$. Figure 14(b) corresponds to one of

the outliers described above in Figure 13(a), where the thresholding technique exhibits a strong bias in its estimate because the

magnitude of the interference spike is high enough to exceed the threshold but not high enough to be flagged as a solid return.

Figure 14(c) is a case of a non-stationary solid return as evidence by the strong peak just above 1 m/s. The thresholding

technique is again strongly biased, the advanced filtering technique produces a valid estimate, and the machine learning

technique gives an estimate that does not meet its confidence threshold, which is not surprising since the technique was not

trained on non-stationary solid returns. While it is unknown what moving object was present within the lidar beam path for

this particular return, non-stationary solid returns are common when scanning nearby rotating turbines. Figure 14(d) is a

difficult case where the advanced filtering technique does not give an estimate because the data are obscured by ground returns,

and the solid return mask, applied when moving between Figure 3(a) and (b), removes the RoI signal, although the machine

learning technique produces an approximately correct estimate. Evidenced by the last two subfigures, there is still development

work to improve the advanced filtering and machine learning techniques for cases with the combination of low velocity and

solid interference.

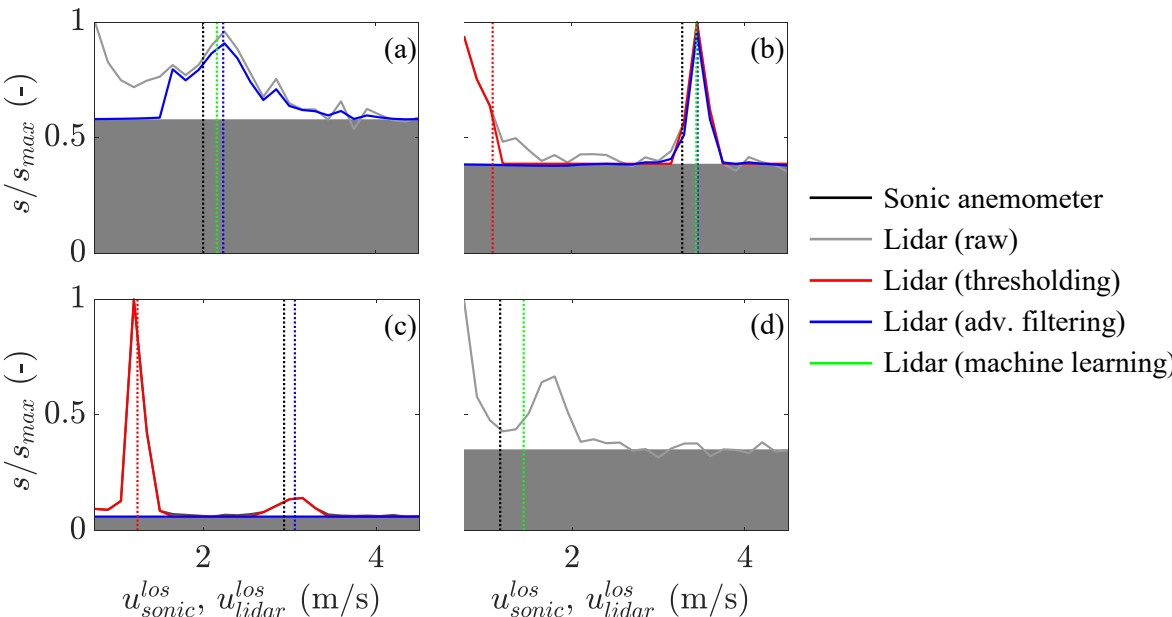

**Figure 14.** Selected examples of scaled lidar spectra, $s/s_{max}$, where $s_{max} = 2^{16} - 1$ versus lidar line-of-sight velocity, $u_{lidar}^{los}$, and sonic anemometer line-of-sight velocity, $u_{sonic}^{los}$, for inflow cases with solid interference. Solid lines ($-$) indicate spectral magnitude and vertical dashed lines (--) indicate the line-of-sight velocity estimates. See text for details.




### 4.2.3. Practical Significance

The error trends reviewed above for inflow cases have implications for wind turbine applications featuring nacelle-mounted, forward-facing lidar. Figure 15 shows the aggregate (i.e., both with and without solid interference) error results as a function of height relative to the wind turbine rotor of the current dataset. Typically, the most important information about the inflow

from a wind turbine control perspective is the flow within the swept area of the rotor, which will be examined below.

The mean errors within the rotor height are small at less than 0.08 m/s and consistent between the three processing techniques as shown in Figure 15(a). The standard deviation of the errors within the rotor height as shown in Figure 15(b), however, are between 0.23 and 0.29 m/s for the advanced filtering and machine learning techniques, and the thresholding technique has 0.002-0.05 m/s (1-22%) higher values depending on scan position because of its poor handling of solid returns.

In terms of aggregated data availability, the advanced filtering technique has 99.7% availability, followed by the machine learning technique at 96.2% and the thresholding technique at 95.5%. Between the two better performing techniques, the advanced filtering is overall more effective than the machine learning within the bounds of the data considered in this study because of higher data availability and slightly better noise rejection.

In practice, the value of the uncertainties quoted above will be increased by reprojection onto the wind direction (i.e., the

reverse process of Eq. (3)). Assuming the lidar center axis is aligned with the wind direction and allowing $\delta = \pm 11.7°$ (which corresponds to the vertical limits of the rotor height in this study for the given focus distance), the $1/\cos(\delta)$ correction will be no more than an increase of 2%, which does not change any of the values quoted in the previous paragraph by more than

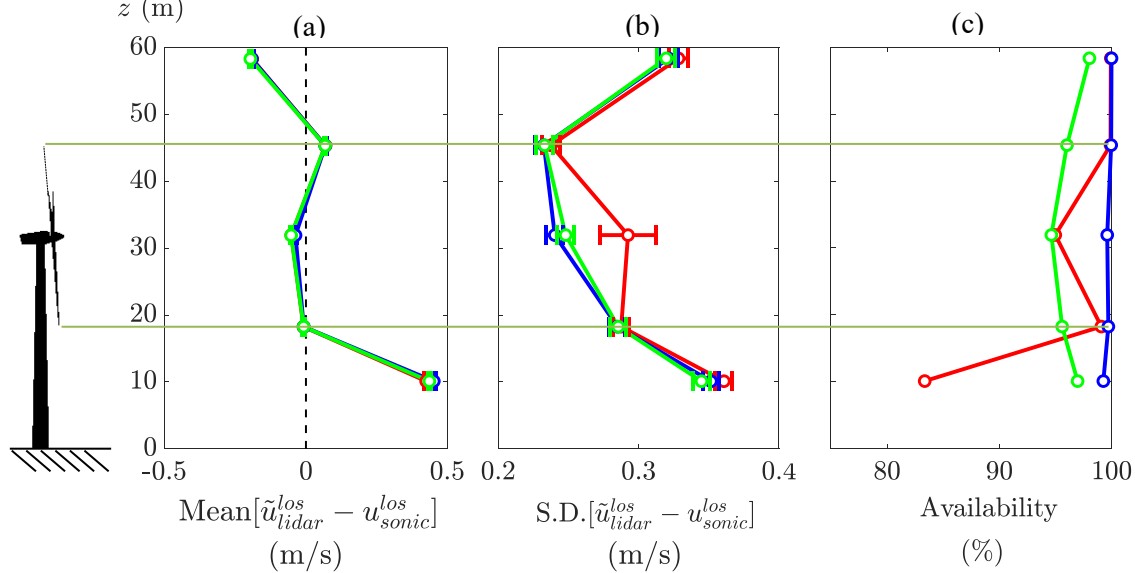

**Figure 15. Aggregate results from inflow cases for (a) mean error, (b), standard deviation (S.D.) of error, and (c) data availability plotted versus height off the ground. The dimensions of a V27 turbine as used in the experiment are shown for reference. The line colors are: thresholding (−), advanced filtering (−), and machine learning (−).**





0.01 m/s. Considering instead a $\pm 30°$ limit on the deviation of the line of sight from the wind direction (which corresponds to the outer cone angle of the DTU SpinnerLidar in this study), the correction will be an increase of $\leq 15\%$. Further increase in uncertainty is introduced by the assumptions typically required about the local wind direction when inferring a velocity from a single line of sight, and these directional biases scale on (1) the RMS magnitude of the transverse (i.e., $y$-$z$ plane) wind components at the scan perimeter of interest and (2) the tangent of the deviation of the line of sight angle from the mean wind direction (Simley et al., 2014).

### 4.3. Waked Cases

This section contains the results from our analysis of the six bins with waked cases described in Table 2. Below, we forego the bulk of the analysis of error trends performed in the previous section, primarily because the smaller sample size of waked cases does not permit a strong study. Rather, we show results that, despite the relatively large error bars on the data, hint at the practical significance of the lidar errors and variations between processing techniques for rear-mounted lidars on wind turbines.

Table 5, Table 6, and Figure 16 are analogous to Table 3, Table 4, and Figure 15, respectively, but for the waked rather than inflow cases. Again, the ranking of efficacy of the three processing techniques (from highest to lowest) is advanced filtering, machine learning, and thresholding, and again the standard deviations of errors are substantially larger than the mean errors. For the advanced filtering and machine learning techniques, the ranges of the standard deviation of errors for cases within the rotor height and without solid interference are 0.29 to 0.45 m/s, and these increase to 0.34 to 0.49 m/s for cases with solid interference. The increase in the upper bound of the standard deviations compared to the inflow cases is expected since the wake presents a relatively turbulent environment, which works against the precision of the lidar in comparison to a point measurement from a sonic anemometer as demonstrated by Figure 12(a). As shown in Figure 16, both the thresholding and machine learning techniques have significantly lower data availability in the waked cases than in the previous inflow cases (note the difference in the horizontal axis limits between Figure 15 and Figure 16), which is related to a higher proportion of solid returns from the meteorological tower in the waked dataset due to the less control that was asserted on the yaw position of the turbine for the waked cases.

The previously mentioned increase in uncertainty introduced by the assumptions required about the local wind direction have been quantified for the waked case specifically in Kelley et al. (2018), who simulated the SpinnerLidar with a $3D$ focus length in a turbulent wake at SWiFT using large eddy simulation. They found an additional mean error after projection on the order of 3% due to deviations of the wake velocity direction from the nominal flow direction. Considering this increase in mean error, as well as again the maximum of 2% increase to all errors due to reprojection onto the wind direction from $\delta = \pm 11.7°$, the maximum mean and standard deviation of error for the aggregated waked cases within the rotor height can be estimated at 0.13 and 0.45 m/s, respectively, for the advanced filtering technique and at 0.17 and 0.47 m/s, respectively, for the machine learning technique.




As noted, the sample size of six bins is too small for a complete analysis. Furthermore, the spatial inhomogeneity of a
waked flow adds further uncertainty to the results since a full analysis should ideally be blocked to account for differences in
processing performance at different points of interest within the wake such as the shear layer and hub flow regions, for instance,
and at different turbine thrust conditions. A more exhaustive dataset is needed and will be sought in future work.

**Table 5.** Performance of lidar processing techniques versus sonic anemometer for waked cases without solid interference. The processing abbreviations are threshold ($th.$), advanced filter ($a.f.$), and machine learning ($m.l.$). S.D. refers to standard deviation.

| Height (m) | | Mean [ $\tilde{u}_{lidar}^{los} - u_{sonic}^{los}$ ] (m/s) | | | S.D. [ $\tilde{u}_{lidar}^{los} - u_{sonic}^{los}$ ] (m/s) | | | Availability (-) | | |
|---|---|---|---|---|---|---|---|---|---|---|
| | | *th.* | *a.f.* | *m.l.* | *th.* | *a.f.* | *m.l.* | *th.* | *a.f.* | *m.l.* |
| 10 | | N/A | N/A | N/A | N/A | N/A | N/A | N/A | N/A | N/A |
| 18 | | -0.03 ±0.05 | -0.01 ±0.05 | -0.01 ±0.05 | 0.30 ±0.04 | 0.30 ±0.04 | 0.29 ±0.04 | 99% | 100% | 100% |
| 32 | | 0.08 ±0.04 | 0.12 ±0.04 | 0.15 ±0.05 | 0.34 ±0.06 | 0.36 ±0.05 | 0.40 ±0.07 | 100% | 100% | 93% |
| 45 | | -0.03 ±0.05 | -0.02 ±0.04 | 0.03 ±0.05 | 0.46 ±0.06 | 0.42 ±0.05 | 0.45 ±0.04 | 100% | 100% | 96% |
| 58 | | 0.07 ±0.03 | 0.07 ±0.03 | 0.07 ±0.02 | 0.24 ±0.03 | 0.24 ±0.03 | 0.23 ±0.03 | 100% | 100% | 100% |
| Combined | | 0.03 ±0.02 | 0.04 ±0.02 | 0.06 ±0.02 | 0.37 ±0.03 | 0.35 ±0.02 | 0.37 ±0.03 | 100% | 100% | 97% |

**Table 6.** Performance of lidar processing techniques versus sonic anemometer for waked cases with solid interference. The processing abbreviations are threshold ($th.$), advanced filter ($a.f.$), and machine learning ($m.l.$). S.D. refers to standard deviation.

| Height (m) | | Mean [ $\tilde{u}_{lidar}^{los} - u_{sonic}^{los}$ ] (m/s) | | | S.D. [ $\tilde{u}_{lidar}^{los} - u_{sonic}^{los}$ ] (m/s) | | | Availability (-) | | |
|---|---|---|---|---|---|---|---|---|---|---|
| | | *th.* | *a.f.* | *m.l.* | *th.* | *a.f.* | *m.l.* | *th.* | *a.f.* | *m.l.* |
| 10 | | -0.08 ±0.27 | -0.26 ±0.03 | -0.18 ±0.02 | 0.26 ±0.22 | 0.36 ±0.04 | 0.27 ±0.03 | 0% | 100% | 79% |
| 18 | | -0.02 ±0.06 | 0.04 ±0.03 | 0.05 ±0.04 | 0.36 ±0.06 | 0.34 ±0.03 | 0.36 ±0.04 | 36% | 100% | 86% |
| 32 | | 0.27 ±0.29 | 0.13 ±0.06 | 0.19 ±0.08 | 0.92 ±0.92 | 0.38 ±0.06 | 0.41 ±0.10 | 21% | 100% | 55% |
| 45 | | 0.11 ±0.10 | 0.18 ±0.10 | 0.21 ±0.11 | 0.34 ±0.14 | 0.48 ±0.08 | 0.49 ±0.09 | 40% | 100% | 74% |
| 58 | | -0.14 ±0.08 | -0.14 ±0.06 | -0.13 ±0.07 | 0.27 ±0.07 | 0.32 ±0.06 | 0.33 ±0.07 | 43% | 100% | 90% |





| Combined | | 0.02 ±0.06 | -0.10 ±0.02 | -0.06 ±0.02 | 0.48 ±0.24 | 0.40 ±0.03 | 0.36 ±0.02 | 17% | 100% | 78% |
|---|---|---|---|---|---|---|---|---|---|---|


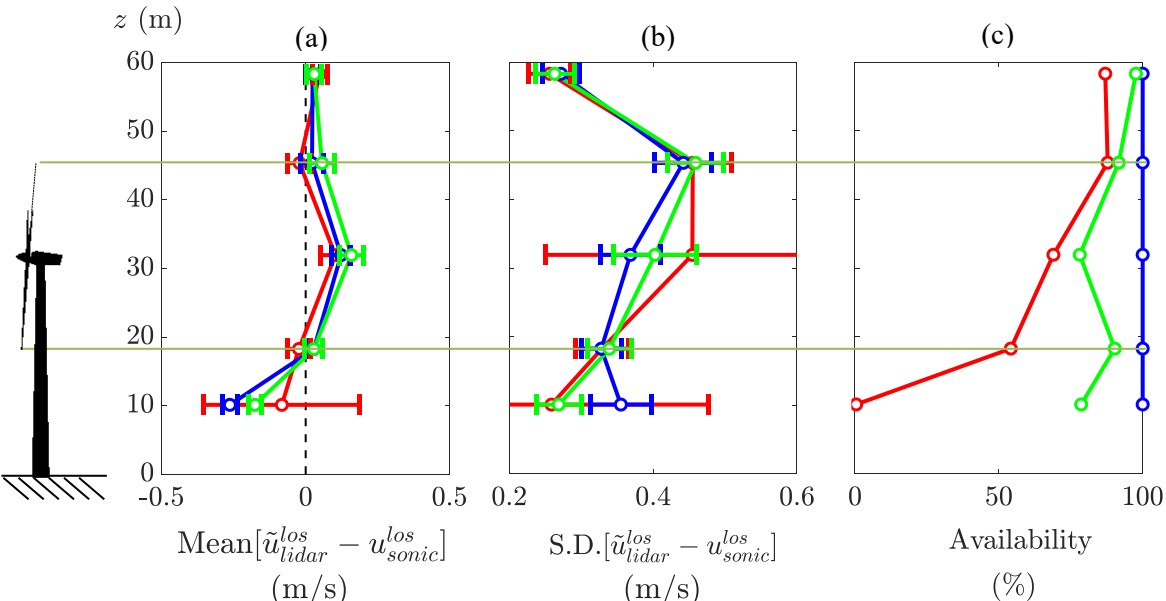

**Figure 16. Aggregate results from waked cases for (a) mean error, (b), standard deviation (S.D.) of error, and (c) data availability plotted versus height off the ground. The dimensions of a V27 turbine as used in the experiment are shown for reference. The line colors are: thresholding (–), advanced filtering (–), and machine learning (–).**

**5.Discussion**

All three lidar processing techniques can produce similar performance when no solid interference is present (assuming the
machine learning model is not exposed to out-of-distribution samples). However, the advanced filtering and machine learning
approaches generally give better performance than the thresholding technique when solid interference is present.

While the thresholding technique's merit for use within the rotor height in this study may be unfairly penalized by the setup
of our validation study that features lidar scans directly over the solid obstruction of the meteorological tower, we note that
inflow lidar scans in the field may, in fact, be required near existing meteorological towers in order to produce higher spatial
resolution of the incoming flow. Other cases that are not considered in the current datasets for which the ability to effectively
reject non-aerosol returns is important are for interference from the optical window (i.e., boresight interference), nearby
turbines, and precipitation. Nearby turbines can present a particularly complicated case because of non-stationary rotor blades,
which shift the solid interference peak away from the typically assumed location at zero velocity. The much poorer





performance of the thresholding technique observed above for all kinds of cases with solid interference represents a main
motivation for using higher-fidelity techniques.

Comparing the advanced filtering and machine learning approaches, a slight performance advantage was demonstrated by
the former for the data considered in our study. However, the potential for improvement of the machine learning technique
may be higher. The advanced filtering technique, which required significant investment during development by subject matter
experts, still requires continuing development for out-of-distribution cases as shown by the outliers and data loss in Figure 13,
for instance. On the other hand, continuing development can be accommodated with relatively low (computational) expense
for the machine learning technique by increasing the size of the networks and/or the diversity of training cases, and this process
requires less expert involvement. The machine learning approach therefore removes the ongoing expert commitment to the
QA/QC problem, instead shifting the workload to a computer. Further, the machine learning technique does not truncate returns
that fall at the beginning of the spectra but implicitly accounts for these (as well as potentially negative) velocities in the QoI
estimation process. The machine learning approach thus provides the framework for both a more efficient workflow
development *and* higher accuracy than is possible by a series of advanced user-generated filters.

Another advantage of the ensemble machine learning approach is the inherent ability to estimate an uncertainty associated
with each estimate. While this feature is employed rather simplistically in this study by providing just a confidence threshold,
the ensemble approach could enable more rigorous uncertainty quantification by leveraging other spectral estimators such as
the higher-order moments and spectral entropy of the distribution of estimates from the individual members of the ensemble.
Other machine learning approaches with uncertainty quantification capability such as the mean-variance, Monte-Carlo
dropout, and Bayesian ones could also be tested. It is cautioned that the machine learning technique should not generally be
trusted to correctly predict confidence on cases that are out of bounds of its training data though so-called *out-of-distribution
detection* offers an avenue for improvement.

A final consideration is computational efficiency. The machine learning technique requires ~1 second to evaluate an
estimate on a personal computer, as compared to ~50 seconds for the advanced filtering technique, which makes the former
more feasible in its current state for real-time control applications.

## 6. Conclusions and Future Work

Three QA/QC processing techniques suited for wind turbine nacelle-mounted lidar were compared and validated using field
data including both inflow and waked cases. The validation study was performed against point measurements from sonic
anemometers mounted to a meteorological tower. Using such a setup, some level of mean and random error is unavoidable
due to flow inhomogeneity coupled with the difference in sample volumes of the two techniques. However, the processing
techniques worked to mitigate uncertainty due to two other sources: amplitude noise and solid interference. Most of the analysis
was performed on inflow cases due to the larger sample size and thus higher statistical confidence in the results. In terms of
mean errors for these inflow cases, the three lidar techniques performed similarly and showed less than 0.08 m/s deviation



from the sonic anemometer data. In terms of the standard deviation of errors, the advanced filtering and machine learning techniques, which showed aggregate errors within the rotor height between 0.2 and 0.3 m/s, performed 1-22% better than the conventional thresholding technique, which could not always filter out solid object returns coming from the meteorological tower and/or ground surface. In terms of aggregated data availability, the advanced filtering technique had 99.7% availability,

followed by the machine learning technique at 96.2% and the thresholding technique at 95.5%. When no solid interference was present, all techniques performed similarly, showing the expected convergence of error with turbulence magnitude and $CNR$. For the waked cases, aggregate standard deviations of errors were larger and on the order of 0.3 to 0.5 m/s within the rotor height for the advanced filtering and machine learning techniques, though this may be attributable in part to a relatively large number of solid returns in the data analyzed due to the experimental setup. The error values above increase by at most

15% due to projection correction based on a maximum of 30° deviation of the line of sight from the wind direction (which corresponds to the outer cone angle of the DTU SpinnerLidar in this study), and small additional errors are introduced especially for the waked cases due to uncertainty in the wind direction.

The machine learning technique showed promise as an approach capable of providing not only less expensive development for QA/QC applications but also higher accuracy and faster evaluation. Future work may include expansion of the parametric

training database to include a wider range of realistic spectral return shapes or development of the workflow to train the model directly on experimental returns sampled near the sonic anemometers, as well as refinement of the machine learning technique to improve confidence levels for each velocity estimate.

New work may also be aimed at validating the estimates of spectral standard deviation of each lidar return, since the inherent spectral width of a volume-averaged lidar measurement may allow for derivation of small-scale turbulence

information. Towards these ends, exploratory work not presented here has shown the advanced filtering and machine learning techniques to be most capable to accurately locate the tails of the lidar spectrum.

## Appendix A: Generation of Synthetic Spectra

The source of truth during the machine learning training process is synthetic spectra with known statistics. This appendix describes the creation of idealized power spectral densities (PSDs) for this purpose. For clarity, we here drop the notation of

$los$ in the superscript of $u$ that has been used to denote line-of-sight velocities above.

The synthetic PSD of the RoI, $s_{ROI}$, is generated as a function of $u$ from a scaled epsilon-skew-normal distribution (Mudholkar and Hutson, 2000) as in Eq. (A-1):

$$s_{ROI} = m_{0_{ROI}} e^{-\frac{1}{2}\left(\frac{u - m_{1_{ROI}}}{m_{2_{ROI}}(1 \mp m_{3_{ROI}})}\right)^2}, \tag{A-1}$$

where $m_{0_{ROI}}$ is a magnitude parameter in counts of 16-bit dynamic range, $m_{1_{ROI}}$ is a location parameter in m/s, $m_{2_{ROI}}$ is a width parameter in m/s, and $m_{3_{ROI}}$ is a nondimensional skew parameter whose absolute value is less than one, and the $\mp$ takes


the sign opposite of the numerator of the exponent. Note that when numerically calculating the true values of the QoIs including

the spectral median from $s_{ROI}$, our discretization extends below 0.75 m/s in order to preclude any truncation of $s_{ROI}$ by the

first several bins that are removed from the spectra as described in Section 3.3. Note that only single-peaked spectra (i.e., no

double-peaked spectra often found at the shear layer of a wind turbine wake) are included in the synthetic dataset since Eq.

(A-1) generates only single peaks.

The synthetic PSD of the solid interference, $s_{solid}$, is generated as an inverse function of $u$ according to Eq. (A-2):

$$s_{solid} = \frac{p_{solid}}{1+(u-u_{solid})/w_{solid}} \; , \tag{A-2}$$

where $p_{solid}$ is the prominence of the solid interference spike, $u_{solid}$ is the velocity at $p_{solid}$ (which in our case is the minimum

$u$ of 0.75 m/s that can be sensed by the lidar), and $w_{solid}$ is the full-width half-maximum of the interference spectrum.

Modeling of the PSD must also include amplitude noise that adheres to the probability density function of the measured

noise content. The statistics of the noise within each PSD bin are known to follow a scaled chi-squared distribution (Rye and

Hardesty, 1993; Garber, 1993). By the central limit theorem, the chi-squared distribution asymptotes to a Gaussian distribution

for sufficiently large sample sizes such as for the hundreds of individual spectra that are averaged in typical lidar measurements.

In such cases, randomized instances of the time-averaged noise spectrum, $s_{noise}$, can be generated given a standard deviation,

$\sigma_{noise}$, and mean, $\mu_{noise}$, within each spectral bin. These noise parameters are here taken to be uniform over the spectrum (i.e.,

we assume white noise).

The combined synthetic PSD, $s$, is constructed in Eq. (A-3):

$$s = s_{ROI} + s_{solid} + s_{noise} \; . \tag{A-3}$$

Collectively, there are eight unknown parameters implicit to Eq. (A-3) including four from Eq. (A-1), two from Eq. (A-2),

and two from the noise contribution. In practice, we retain only seven of these parameters; the only independent noise

parameter is $\sigma_{noise}$ because $\mu_{noise}$ is determined by the scaling approach described next.

To mimic the scaled version of the SpinnerLidar data output (see Branlard et al., 2013), a vertical translation is applied to

each $s$ curve so that the maximum value of the curve is the full magnitude of the 16-bit SpinnerLidar output. Depending on

the maximum prominence of $s$, this translation effectively sets $\mu_{noise}$.

For each of the seven parameters, a full-factorial sweep across a range of values that have been observed in measurements

provided a database of lidar spectra to train and test the machine learning model. In order to ensure that this parametric space

be representative of the true population of observed lidar spectral shapes (notwithstanding the limitations due to using only

single-peaked spectra), the range of variation of the parameters was drawn from 3.2e8 individual lidar returns taken over more

than 180 hours of wake sampling, which included both daytime and nighttime conditions, as well as the winter, spring, and

summer seasons. These results, which included lidar returns at focus lengths of 1.0$D$, 1.5$D$, 2.0$D$, 2.5$D$, 3.0$D$, 4.0$D$, and 5.0$D$,





represent the full dataset from the experiments at the SWiFT site during the 2016-2017 wake steering campaign (2019). Note that this dataset is substantially larger than the selection of bins described in Section 3.4.1.

Table A-1 shows the approximate ranges of parameters extracted from the full dataset. In several cases, the range listed in the table (and used for model development) is reduced slightly from the extracted values to lessen the complexity of the training dataset given the limited number of training cases to be generated. Most notably, $m_{1_{ROI}}$ has a minimum of 2 m/s since the interaction between the RoI and the solid interference signature is increasingly difficult for the machine learning model as these two regions converge.

Within the ranges given in Table A-1, the full factorial synthetic data are generated from four uniform or logarithmically-spaced intervals across the range depending on the parameter (as noted in the main text, our synthetic dataset matches the ranges of the observed parameters but not yet the distributions). This results in 78,125 training cases, and the training process was found to be more robust if parameter values for each case were randomized within their interval. An additional 16,741 validation cases and 16,741 testing cases were generated using a uniform random distribution over the parametric space, and

these cases were used to determine when to terminate model refinement and to test the final ensemble predictions, respectively, as described in Section 2.3.2. Before initiating the training process, the training, validation, and testing data are filtered based on a requirement that $m_{0_{ROI}} > 4\sigma_{noise}$, which reduces the number of synthetic cases to 58407, 13428, and 13383, respectively (a field implementation of the current machine learning technique could feature a preprocessing thresholding operation to flag and/or reject cases of lowest $CNR$ that fall outside this bound of the training data).


**Table A-1**. Ranges of parameters used for generation of synthetic spectral dataset.

|  | $m_{0_{ROI}}$ | $m_{1_{ROI}}$ | $m_{2_{ROI}}$ | $m_{3_{ROI}}$ | $p_{solid}$ | $w_{solid}$ | $\sigma_{noise}$ |
|---|---|---|---|---|---|---|---|
| Minimum | 1.2E+03 | 2.0E+00 | 7.5E-02 | -2.5E+00 | 0 | 1.2E-01 | 5.1E-01 |
| Maximum | 4.0E+04 | 1.3E+01 | 1.0E+00 | 2.5E+00 | 6.0E+04 | 6.0E+00 | 3.2E+03 |

*Data Availability.* The data can be made available upon request.

*Author Contributions.* Kenneth Brown was responsible for the development of the machine learning technique, data post-processing, and manuscript. Thomas Herges led the lidar measurement campaign, developed the advanced filtering technique, and contributed to the manuscript.

*Competing Interests.* The authors declare no conflict of interest.






*Acknowledgements.* The authors gratefully acknowledge Torben Mikkelsen and the team at the Technical University of Denmark (DTU) for their collaboration surrounding the SpinnerLidar. Dan Houck and Christopher Kelley are thanked for their internal review of this manuscript and discussion thereafter. David Maniaci is also thanked for his guidance on the direction of the research project and manuscript.


*Financial Support.* This research was supported by the Wind Energy Technologies Office within the U.S. Department of Energy's Office of Energy Efficiency & Renewable Energy (see below for contract details).

Sandia National Laboratories is a multimission laboratory managed and operated by National Technology & Engineering Solutions of Sandia, LLC, a wholly owned subsidiary of Honeywell International Inc., for the U.S. Department of Energy's

National Nuclear Security Administration under contract DE-NA0003525. This paper describes objective technical results and analysis. Any subjective views or opinions that might be expressed in the paper do not necessarily represent the views of the U.S. Department of Energy or the United States Government.

*Review Statement.* This paper was edited by <editor name> and reviewed by <reviewer name or anonymous> and <reviewer

name or anonymous>.

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
