# Peer review of "High-Fidelity Retrieval from Instantaneous Line-of-Sight Returns of Nacelle-Mounted Lidar including Supervised Machine Learning"

_Atmospheric Measurement Techniques, 2022_

## Author Comment (AC1)

Monday, September 19, 2022

Dear Referee Number 1:

Thank you for the very detailed and helpful comments on our article. Please see below our responses in green to the items you mentioned to be revised.

General Comments
The authors present an investigation which focusses on using machine learning to improve
the processing of Doppler spectra acquired by scanning Doppler lidars. The manuscript is
both well structured and written, which enables the understanding of the processing steps
used in the study as well as the rationale for following them. Their results show that
through the use of machine learning it is possible to increase the data availability
especially in the case when a scanning Doppler lidar is measuring complex flows, like a
wake of a wind turbine. In those cases, the differentiation between wind signals and noise
is a challenging task and the conclusions of this study will contribute in addressing these
challenges.

Please find below my comments on the manuscript, as well as suggestions for
improvements that I think will enhance the understanding of the presented study.

- The authors throughout the manuscript discuss about the interferences that the
  optical window of the scanning wind lidar is creating in the measured Doppler
  Spectra. I think that rather a general problem for wind lidars, this issue is a
  characteristic of the scanning wind lidar used in this study. It is probably
  attributed to alignment of the top window to the line-of-sight of the lidar. This can
  be dealt if the top window is not installed normal to the line-of-sight of the lidar,
  as it happens for example in the commercial ZX300 wind profiler. I suggest that the
  authors state that this is relevant for this instrument or provide a reference that
  this is a general problem. Furthermore, the near zero velocity contributions could be
  suppressed using an electrical high-pass filter. We agree with the referee's comment
  and believe we have addressed the problem. We have added mention in multiple places
  that the interference from the optical window is specific to the SpinnerLidar
  configuration rather than a general problem amongst lidars. The first mention in
  Section 1 now states, "For nacelle-mounted lidar, such interference is common and
  stems from the surrounding terrain, optical windows (i.e., boresight interference,
  which occurs near the center of the field of view for the SpinnerLidar configuration
  (Brown and Herges, 2020) when the line-of-sight is normal to the weatherization
  window), neighboring turbines, …". The mention of "boresight interference" in Section
  2.2 is made in the context of the setup at the SWiFT site using the SpinnerLidar, so
  we have left it unchanged. The mention of "boresight interference" in Section 5 has
  been modified to read, "…interference from the optical window (i.e., the boresight
  interference that affects SpinnerLidar measurements)…". Your comment about the high-
  pass filter seems also to be a good one, though this is not an area of expertise for
  the authors.
- In Figures 11 and 13 are presented scatter plots of the line-of-sight measurements
  from the lidar and sonic anemometers. The lidar data used correspond to
  quasiinstantaneous measurements, acquired at 400 Hz, and the sonic anemometer data

that was at 100 Hz "has been temporally interpolated at the same instance as the lidar passing". In those high sampling rates, it is very important to ensure a very good synchronization between the two data acquisition systems. Can the authors add information about the expected synchronization accuracy? This is a great point, and we have added some details. Firstly, in Section 2.2 where the ultrasonic anemometers are introduced, we added "There is an estimated 25–30 ms delay from the end of each sample until when the GPS timestamp is applied (i.e., ~20 ms internal delay in the instrument and 6–7 ms serial delay)." In the following section that describes the SpinnerLidar, we added, "The delay time between sample and GPS timestamp is less than 1 μs". These details get referenced again and tied together in Section 2.4.3, "It is noted that the above procedure of comparing measurements between a stationary sensor and a non-stationary sensor requires good temporal synchronization. The synchronization is accomplished with GPS timestamps on all sensors. The synchronization accuracy then becomes a function of any delay occurring between the data-capturing and time-stamping processes, especially for the non-stationary sensor. For the non-stationary SpinnerLidar, the delay of <1 μs introduces negligible error in the perceived position of the beam as it is more than three orders of magnitude smaller than the sample period of each individual measurement location in the rosette pattern. For the stationary ultrasonic anemometers, the delay of 25–30 ms is small compared to the timescales of the flow being resolved by the 8.45 m lidar probe volume."

- The authors in the lines 197 – 198 mention the wind turbines WTGa1 and WTGa2, but the presentation of the experimental setup in first presented in Figure 7. I suggest that the authors should consider moving the presentation of the Experimental Techniques section after the Introduction, so it is easier for a reader to understand the discussion about the processing techniques. Yes, we agree with your point and have made the switch.

- The authors in Section 2.2 give an overview of the "Advanced Filtering" technique. I find the overview quite extended. Maybe the authors could consider highlighting the differences in relation to the method presented in Herges and Keyanto (2019) and reduce a bit this part. This point is well taken, and we have reduced the length of Section 2.2 by almost a page including eliminating the most "in the weeds" discussion and figure surrounding the final filtering step. As the text indicates, further details of the filtering formulation can be found in Heges and Keyanto (2019). This reduction brings the total length of Section 2.2 to be similar to that of Section 2.3, which we see as appropriate since the article serves equally as a validation of the "advanced filtering" technique as well as the "machine learning" one.

Specific Comments
- Line 46. Can the authors elaborate more what they mean with "typical setups"? "Typical setups" was intended to refer to the setups of the majority of commercial and research lidar setups, similar to the two references later in the sentence. We've updated the phrase to "typical commercial and research lidar setups".
- Line 56. Here the authors talk about the "time series of u_los", thus, if I understand this correctly, referring to the line-of-sight velocity that is derived by the estimated characteristic frequency shift in each Doppler spectrum. However, in the caption of Figure 1 they use "u_los" to also denote the various line-of-sight contributions over a frequency bandwidth of the region of interest. I find this a bit confusing, thus I suggest that the authors clarify this part. For example, they could

use the same symbol as the one they use for the median line-of-sight. This is a good point; the nomenclature is being confused here. We want to wait to introduce the specific symbol for median line-of-sight velocity and instead be more general at this point in the text since the spectral broadening will affect multiple processed quantities of interest. We have changed the text to read "processed quantities of interest". Please let us know if this reads more accurately.

- Lines 68 – 72. Here the authors describe noise contributions in a laser Doppler spectrum. First, I suggest that they should mention here that this is relevant to a cw Doppler lidar. Furthermore, I think that this description is not accurate: a. The presence of a flat noise floor doesn't result to a loss of precision of the line-of-sight velocity estimation, since it can be effective removed by thresholding. The mean noise floor due to shot noise is dependent on the intensity of the local oscillator laser and not on the backscatter. b. The SpinnerLidar uses a coherent detection scheme which is based on a balanced photodetector, therefore I don't see how is relevant the reference of Liu et. al 2006, c. The variance of the noise floor, which is dependent on the number of FFTs averaged to produce a Doppler spectrum will introduce a noise in the measured Doppler spectrum, the impact of which will be dependent on the intensity of the backscattered light, the frequency bandwidth of the wind speed fluctuations, and the method used to estimate the characteristic Doppler shift. We agree with your criticisms here and have made some tweaks to be more truthful. First, the introductory sentence of the paragraph now includes a note to indicate that this discussion is related to "CW Doppler lidar". Also, we have removed altogether the note about the noise floor since, as you describe, the presence of the noise floor neither affects the precision of the measurement nor is a result of the intensity of the backscatter that is a main part of the narrative in this paragraph. We have added that the intensity of the noise depends "in part" on the range-resolved intensity of the backscatter since there are other considerations, as well. Finally, the inappropriate reference to Liu et al. (2006) has been removed.
- Line 86. Rather than an increase in the signal strength, the accumulation of a number of spectra reduces the noise variance. Good point. We have changed the text accordingly.
- Lines 154 – 156. Can the authors explain this part a bit better? I am not that I understand the statement "...this magnitude is the maximum among the bins...". We have added a parenthetical explanation: "this magnitude is the maximum among the bins (i.e., the maximum over all $u_{los}$)". Please let us know if that is still not clear.
- Lines 168-169. Can the authors explain what they mean when they write that they "don't apply any correction for the alteration of the skew distribution due to thresholding"? By this, we are referring to the fact that chopping off the bottom portion of a skewed distribution introduces a slight bias into the mean frequency estimation. We are not applying any correction for that bias after our thresholding, but it should be small. We have removed this sentence altogether since it may be unessential.
- Line 184. What kind of scan-head motor speed rates did the authors test? We tested with three scan-head motor speed rates: 500, 1000, 2000 rpm. We have added these values to the text and also, for consistency, added the range of focus distances tested, "throughout all focus distances (1.0$D$, 1.5$D$, 2.0$D$, 2.5$D$, 3.0$D$, 4.0$D$, and 5.0$D$) and scan-head motor speed rates (500, 1000, 2000 rpm)."
- Line 199. In Figure 2c the authors present a time series of normalized Doppler spectra. For the normalization they subtract the mean noise floor from the measured

spectra and subsequently divide with a fixed value. I don't understand the point of this division, it is more a rescaling of the spectrum rather a normalization. It is a good observation that this is a very liberal use of the word "normalization". We have removed this word and replaced it with "noise-subtracted and rescaled" in all six places where it was mentioned. The point of the division is just to make the plot axis more convenient to interpret.

- Line 205. The authors state that "the mask regions were then increased horizontally by 2 pixels". What is it meant here with both "horizontally" and "2 pixels"? We have modified this to say, "the mask regions were then increased to include two additional scan indices in both directions".

- Line 229. How large is the area of the sliding neighbourhood and how did the authors reach to that value? This value is a certain factor smaller than the area of the scan pattern and was chosen so that there would be a sufficient number of points in the neighborhood at any position in the scan pattern. Sufficient number is defined as being wide enough to capture "crossings" of the rosette scan pattern while excluding capturing the majority of the adjacent rosette branch. This provides a way of smoothing velocity and capturing outliers relative to time differences in the rosette branches. To address the referee's fourth General Comment above, we are removing some of the discussion surrounding the sliding neighborhood since this maybe too fine of detail as the referee suggests. However, more information on the sliding neighborhood is in the Herges and Keyanto (2019) reference cited at the beginning of the section.

- Line 234. The authors state that in Figure 4b the "peak returns signals from the operational rotor are clear". However, I am not sure if I can visually detect them. Can you please explain a bit more in the text what are the observations that one should notice in that figure? In response to your fourth General Comment, we have removed this sentence (and in fact all of Figure 4) as it may be more detailed than is necessary considering that Herges and Keyanto (2019) also discuss the method in detail. However, to answer the question, the key to interpreting that figure is the threshold marker in green on the color bar. If the color of a scatter point falls above the green threshold line, it corresponds to a scan that is a peak return outlier.

- Lines 277 and Lines 285. Since, I am not a machine learning expert I don't know how
- typical are the values of B and of the percentages of the data split. Can the authors add a short comment or a reference in the manuscript to support or explain the selection of these values? Related to the value of B, we have added this comment to help justify our selection: "Typical values of B are between 20 and 200 (Tibshirani, 1996); we use B = 32". Related to the percentages of the data split, there is some ambiguity on the ideal split. Hastie et al. (Springer, 2001) is an early and well-cited reference who give a ratio of 50:25:25 but admit that "it is difficult to give a general rule on how much training data is enough; among other things, this depends on the signal-to-noise ratio of the underlying function, and the complexity of the models being fit to the data." Some recent authors have used ratios with a higher percentage of training data such as our 70:15:15 or an even a higher first number.

- Lines 329 – 330. Here the authors the authors write that the selected threshold of the standard error provides an acceptable balance between data availability and variance error. However, they also state that the trade off has not yet been studied exhaustively. On which basis it is concluded then that it provides an acceptable balance? This is a good question. A simple parameter sweep of the threshold level helped identify the final threshold selection based on (approximately) minimizing the

scatter in the synthetic test data error while also retaining most of the synthetic test cases. As indicated, it is a somewhat heuristic selection at this point. A more thorough approach that could be taken in future work might be to sweep through the threshold on the actual experimental data and replot figures 10(c) and 12(c), for instance, for each threshold value. Although it does not offer the reader TOO much, we have improved this line to say: "This threshold provides an acceptable balance between data availability and variance error based on a parameter sweep applied on the synthetic dataset, though the tradeoff has not yet been studied on the experimental dataset."

- Line 371 I suggest replacing the reference of Mikkelsen et al. 2013 with the Sjöholm, M., Pedersen, A. T., Angelou, N., Abari, F. F., Mikkelsen, T., Harris, M., Slinger,C., and Kapp, S.: Full two-dimensional rotor plane inflow measurements by a spinnerintegrated wind lidar, in: European Wind Energy Conference & Exhibition 2013, 2013. Since, in the Mikkelsen et al. 2013 a spinner lidar with a single prism scanner head was used. OK, this is a good point. We made the replacement here in one spot in the Introduction where the Sjoholm et al. reference is more appropriate.

- Line 414. Table 2. The mean wind direction that is reported here does not seem correct. For that wind direction the wake of the wind turbine would not be in the location of the mast. How is the mean direction estimated here? This was a mistake, and we have fixed it. Previously, the routine to calculate a mean direction with correct accounting for the discontinuity at 0/360° was only being applied to the yaw heading and not to the wind direction, as well. This has now been fixed, and the mean wind direction in Table 2 is now 346.81°. The calculation has also been applied for Table 1 although the values in Table 1 remained unchanged since both the wind direction and yaw heading values were tightly clustered away from the 0/360° discontinuity for that table. The reporting of the min and max values across the bins has also been changed to a more intuitive convention.

- Line 441. Please define δ also in the text, not only in the caption of Figure 8. Done.

- Page 21. In the legend of the Figure 9 it is written that the spectra presented in Figures c – g are scaled with a fixed number ($2^{16}-1$). However, just from a visual inspection of the figures, all presented spectra appear to have the same maximum value, which maybe is 1? Are the authors sure that they scale the spectra with a fixed number and not with the maximum value of the measured spectra? it is otherwise strange that all measured spectra have the same maximum value. Your observation is correct. When pulling the data for this analysis, we retrieved the scaled output. This output is described in Branlard et al. (2013) as being used to enable "convenient storage". We added a note at the bottom of the captions for both figures where this point is helpful to explain, "Note that the maximum value of each subfigure is always unity because the authors retrieved the scaled version of the SpinnerLidar output; see Branlard et al. (2013) for context."

- Page 24. Figure 11. In this figure are presented scatter plots between lidar line-of-sight measurements and the corresponding sonic measurements projected on the lidar's lineof-sight. Here it looks like the measurements at 10 m (presented in orange) have overall higher values than the ones at 58 m (presented in red). This is inconsistent with the profile presented in Figure 9. Can the authors explain why is this difference observed? We believe the referee is referring to Figure 13 not Figure 11 (in the new manuscript, this is Figure 12 rather than Figure 10). In the figure, the only three discernable colors are orange, yellow, and purple, which correspond to the

heights of 10 m, 18 m, and 32 m, respectively. Cross-referencing with subfigure (c) in Figure 15 (Figure 14 in the new manuscript), it is evident why this is the case because the data availability of the thresholding technique at the two highest sonic positions is nearly 100%. In our dataset, high data availability of the thresholding technique indicates low prevalence of solid interference. From subfigure (c) of Figure 15 (14) it is also apparent that there are considerably more solid returns from the 10 m position than from the other two positions, which is intuitive since the strongest source of solid interference in our configuration is the ground. The larger number of solid returns at the 10 m position, combined with the fact that the 10 m data is the top layer of the plot, explains why orange dominates the plots in Figure 13 (12). You would still expect to see *some* amount of 18 m and 32 m data above the 10 m data because of the argument the referee is making. Looking closely at subfigure (b), for instance, of Figure 13 (12) you do indeed see a grouping of 32 m data at higher magnitude than the 10 m data. The referee's comment is a good one, and to help clarify for readers, we have added this line near the discussion of Figure 13 (12), "Note that unlike in Figure 10, Figure 12 is dominated by the data from the lowest sonic anemometer position, which follows because the strongest source of solid interference in our configuration is the ground."

- Lines 538 – 543. What is the direction of the booms in relation to the geographic North? The booms are due west (270° in coordinate system of original Error! Reference source not found.) of the tower. This info has been added to Section 3.2. Further, we have added a parenthetical note at these Lines 538–543 related to the nonexistence of direct waking of the sonic anemometers in our validation study, "note that direct waking of the sonic anemometers from the meteorological tower should not be encountered in this study because of the boom orientation and constraint on wind direction mentioned above".

- Line 553. Why do the authors use the letter T to denote standard deviation and not the Greek letter sigma? $\sigma$ was already claimed in the discussion of spectral amplitude noise where the parameter $\sigma_{noise}$ is used (see Sections 2.3.2 and Appendix A).

- Line 557. What is the "parameter estimation problem"? This is terminology that we have seen in some studies related to lidar and more generally in studies involving mean frequency estimation, sometimes referring to the problem of determining quantities like spectral mean and standard deviation from noisy data. Here is a general example: FAN, Z., et al. "Cramer-Rao Lower Bounds of Parameter Estimation for Frequency-Modulated Continuous Wave (CW) Lidar." (2018).

- Line 584. Table 3. Would not it more relevant here to estimate the mean and standard deviation of the absolute difference between lidar and sonic line-of-sight measurements? And where it is attributed the drop in the data availability that is observed in the machine learning technique? The same comments apply to Tables 4, 5 and 6. We understand the referee's comment about using the absolute differences to calculate the statistics. We do think it is valuable to show the direction of the bias when a mean error is present, which is why we showed the data as is. The drop in data availability for the machine learning technique is attributed to the process described in Section 2.3.3. When the standard deviation of the individual neural network outputs exceeds the threshold given, the machine learning estimate is rejected on the basis that the networks cannot agree on the right answer. For cases without solid interference, the threshold is rarely needed as shown by the data availabilities near 100% in Tables 3 and 5, and more machine learning training should be done to bring the data availability to 100%. Earlier in the manuscript during the

discussion of original Figure 11, we have this sentence, "Notably, the data availability for the machine learning technique is 3% lower than for the other two techniques *(see Section 2.3.3. for the explanation)*, though this gap might be helped with an improved machine learning architecture and training scheme." The text in italics has been added based on the referee's comment to help connect the reader back to the cause for this data loss. For cases with solid interference, the estimation of median is a more difficult problem, so we have lower data availabilities in Tables 4 and 6. Again, a more refined training process for the machine learning should help here.

- Page 28. Table 4. Why there are no measurements at 45 m and 58 m in the case of the "thresholding" and "machine learning" techniques? Furthermore, how do the authors explain that even though the machine learning technique can eliminate biases that are attributed to the truncation of the low energy part of the Doppler spectrum the statistical results are still similar with the other techniques? Unlike at the 10 m and 18 m positions where ground interference is prevalent and unlike at the 32 m position where the scan pattern was found to often hit the sonic anemometers and boom themselves, there is not much solid interference to speak of at the 45 m and 58 m positions in our experiment. Thus, there is a quite small sample size at the two higher heights in Table 4. The cases of solid interference that DO exist are full solid returns (i.e., ones where the solid spike is the largest magnitude signal in the spectrum), and for such cases the thresholding technique (by design) produces no estimate while the machine learning technique for these specific cases does not meet the prediction confidence threshold to reliably report data. Related to the referee's second point, we did in fact mention earlier in the article, "While the machine learning technique offers the possibility to eliminate such a bias as noted in Appendix A, we do not observe any practically significant differences in the mean offset or slope of the linear trend lines between processing techniques in the dataset in Figure 11." We have now supplemented that comment with the sentence, "More advanced or exhaustive training may be needed to reap this benefit from the machine learning approach."

- Line 635. The reverse process of Eq. 3 is not straightforward using line-of-sight measurements from one Doppler lidar and it requires assumption of spatial homogeneity of the flow or the use of a model of the spatial variations of the flow. Yes, this is a good point; our language is imprecise. We have removed this parenthetical reference to the "the reverse process of Eq. 3" since we are just referring to simple projection of the line-of-sight back onto the mean wind direction without accounting for lateral velocity components.

- Please explain in which context the correction by the inverse cosine of delta is discussed. We believe the referee is referring to the discussion of the inverse cosine of delta at Line 636 in the original manuscript. The inverse cosine correction would be required to get the uncertainty in the wind speed estimate (in the mean wind direction) based on a line-of-sight measurement (away from the mean wind direction). Of course, there are other uncertainties not captured by a simple inverse cosine correction, and we made note of this at the end of the paragraph, "Further increase in uncertainty is introduced by the assumptions typically required about the local wind direction when inferring a velocity from a single line of sight, and these directional biases scale on (1) the RMS magnitude of the transverse (i.e., y-z plane) wind components at the scan perimeter of interest and (2) the tangent of the

deviation of the line of sight angle from the mean wind direction (Simley et al., 2014)."

- Line 710 – 712. It is stated that the machine learning technique requires 1 second to evaluate an estimate on personal computer. What is the duration of the data set that this processing time corresponds to? We have updated the text to read, "The machine learning technique requires ~1 second to evaluate a full 984-point rosette estimate pattern (i.e., typically 2 seconds of scan time) on a personal computer…".

Technical Corrections
- Line 7. I suggest replacing "error" with "errors" Done.
- Line 15. I suggest replacing "an overlapped" with "an adjacent". I understand that the authors mean that the meteorological tower was at the same location as the scanning pattern, but the adjective "overlapped" without additional information is a bit confusing here. Done.
- Line 33. Can you add here what type of models you are referring to? The models include a range of wake models: steady-state analytical, dynamic wake meandering, Reynolds-averaged Navier-Stokes, and large-eddy simulation models. We were not sure if THAT level of detail is appropriate but have simply added the modifiers, "wake aerodynamics" before the phrase "model validation". Let us know if that seems insufficient.
- Line 36 – 38: Please rephrase this sentence. The uncertainties in lidar measurements cannot stem from the modelling approaches for the reconstruction of the velocity vector. It is rather that uncertainties in the estimated radial wind speeds that propagate to uncertainties in the reconstructed wind vector. Your point is well taken. I believe this sentence is now more precise, "Uncertainties in processed lidar  data stem both from the lidar line-of-sight velocity, $u_{los}$, readings themselves and from  imperfect assumptions in modeling approaches for reconstruction of the velocity vector (Lindelöw-Marsden, 2009; Van Dooren, 2021)."
- Line 41. The spectrum presented in Figure 1 corresponds to a Doppler spectrum that is acquired by a continuous-wave (cw) Doppler lidar. I suggest stating here that this is an example of a cw lidar. Done. Also, we added a mention of "CW" to the caption of Figure 1.
- Line 43. Add "the" to "…are related to the line-of-sight velocity…". Done.
- Figure 1, label. I suggest replacing the "Example power spectral density.." with "Example of a power spectral density…" Done.
- Line 64. Add "the" to "…sources in the measured lidar…" Done.
- Line 66. "…instrument errors …" Done.
- Line 67. "… nacelle-mounted lidars" Done.
- Line 72. "… is a fast scanning …" Done.
- Line 72. ".. CW lidar as have been mounted…" please edit this sentence, it looks that something is missing. Yes, we have changed the sentence to read more clearly, "A particular configuration of interest to our work is a fast-scanning (i.e., ~500 Hz) CW lidar that has been mounted on turbine nacelles (Sjöholm et al., 2013)".
- Line 73. The reference Mikkelsen et al. 2013 refers to a field study with a lidar installed on the spinner of a wind turbine, which did not sample at 500 Hz. We have changed this reference to Sjöholm et al. (2013).
- Lines 192 – 194. The authors write "Line colors correspond between each of the three subfigures…" . Please clarify this sentence. We have change this to, "The different line colors demarcate the seven scan indices of interest…"

- Line 248. Replace "is" with "has". We believe the referee may be referring to a different a line since the mention of "is" in line 248 seems grammatically appropriate.
- Line 262. Add "using the Levenberg–Marquardt method". Done.
- Line 283. Add "the model refinement" Done.
- Line 377. What is meant with the "full weight"? Please see this modification, "Integrating the weighting function over a length of 16 × FWHM centered around this focus length captures over 99% of the area under the full weighting function".
- Line 409. More than how many orders of magnitude? We have updated this to read "more than one order of magnitude".
- Line 411. The units of the values presented in Table 1 and 2 should not be written in italic fonts. OK, this has been fixed.
- Line 448. Remove one of the two section references. Done; thank you.
- Line 451. Add "…on the ensemble…" Done.
- Line 650. Add "than the inflow cases". Done
- Line 721. Add "the three lidar processing techniques" Done

---

## Author Comment (AC2)

Monday, September 19, 2022

Dear Referee Number 2:

Thank you for the well-articulated and helpful comments on our article. Please see below our responses in green to the items you mentioned to be revised.

General Comments
- In contrast to the two other techniques the Machine Learning approach not only filters spectra, but processes the whole way down to u_LOS. It is not really a QA/QC technique anymore but rather a full retrieval, hence I consider statements like "Our work compares three QA/QC techniques, including conventional thresholding, advanced filtering, and a novel application of supervised machine learning ..." (l. 10) as inaccurate. This comment is well taken. In truth, all three methods described in the Processing Techniques section include both QA/QC *and* mean frequency estimation. From that perspective, all the techniques could be considered full retrievals, and we have synced the terminology throughout the article to address this point. Specifically, references to "QA/QC techniques" have been modified to be "processing techniques" except for the several cases where the QA/QC process in particular is being discussed. The first time the term "processing techniques" is introduced, it is now described as "…processing techniques (i.e., full retrievals including both quality assurance/quality control and subsequent parameter estimation)".

Why did you not choose to just using ML for QA/QC to produce filtered spectra and then run the peak detection on these spectra as you do with the other approaches? This would make the approach more general and give more insight on what the model actually does, hence allow for more targeted improvements We agree that your suggestion makes good sense and would have been a possibly more scientifically rigorous approach to this study. Unfortunately, we set the architecture of the ML process early in our work to process all the way to the QoI. We have added a line in the Discussion section that addresses your point, "It is also noted that more targeted improvements to the machine learning technique might be possible if the technique was designed to produce intermediate filtered spectra rather than only estimating the final QoI."

- The paper could make more clear right from the abstract and the introduction that the main aim of the current work is to get rid of solid interferences while keeping the data availability high. Indeed Fig 11 and Tab. 3 show that for the other cases already the very basic thresholding approach is enough to achieve comparable results than for the two more advanced techniques We agree with your point that the primary outcome of this work is to remove solid interference while keeping data availability high. In the third sentence of the abstract, we added the phrase "especially that due to solid interference" to differentiate that there was greater success in reducing error from solid interference than in reducing error from amplitude noise, which was the other source of uncertainty targeted by our work. Further, in the next sentence we added the underlined words: "Our work compares three QA/QC techniques…based on their ability to reduce uncertainty introduced by the two observed non-ideal spectral features while keeping data availability high". We also highlight the data availability aspect with a new clause in the Introduction section, "we compare $\tilde{u}^{los}$ to corresponding values measured from a meteorological tower co-located with the lidar focus point while also tracking data availability associated with the different QA/QC processes.

We also made a change in the Conclusion section where it was added: "However, the processing techniques worked to mitigate uncertainty due to two other sources, amplitude noise and solid interference, while keeping data availability high, and most of the benefit of the higher-fidelity techniques stemmed from the reduction of error from solid interference." We believe these edits highlight the primary outcome of the work while not entirely removing reference to the ability of the processing techniques to sometimes reduce error stemming from amplitude noise, which was demonstrated to a small degree in Figure 11.

Specific Comments

- 15: please re-word "overlapped meteorological tower" to something like "on-site meteorological tower" or "meteorological tower within the sampling volume" We have changed this to "adjacent meteorological tower within the sampling volume".
- Sect 2.2: This description is quite long for that it actually principally follows Herges and Kayantuo, 2019. Please consider a more concise formulation focusing on potential differences to what is presented in the above-mentioned reference. Nevertheless, the illustrations Fig. 2 and 3 are rather valuable, I wouldn't skip these. This point is well taken, and we have reduced the length of this subsection by almost a page including eliminating the most "in the weeds" discussion and figure surrounding the final filtering step. Note that we keep Fig. 2 and 3 while removing Fig. 4. This reduction brings the total length of the subsection to be similar to that of the following subsection on the machine learning technique, which we see as appropriate since the article serves equally as a validation of the advanced filtering technique as well as the machine learning one.
- Fig. 2: I found it a bit confusing to have the most "raw" data in subfigure c) while excerpt spectra of it are in b) and processed median LOS winds in a). If I didn't miss anything fundamental, I would prefer to change c) <-> a) Please see Figure 4 in the new manuscript. We agree with you and have switched the order to what you describe and updated the text and caption accordingly.
- Fig. 3: Please leave a reference to the original spectra in Fig. 2b) in the caption of Fig 2) The authors are not quite sure what the referee is asking. These are the original spectra that came directly out of the instrument save the offset and scaling described by the axis label.
- 301: you deliberately exclude double-peaked spectra for training what is to be expected present for wake situations. Why this choice? How to explain that the Machine Learning results fit the anemometer data that well in Fig 10d)? Double peaks occur only when very steep gradients occur near the focal point. It is not expected to ever see double peaks for inflow cases, which are the focus of our study, because the atmospheric shear is usually not so severe. The referee rightly asks about double peaks in the waked situations. We would indeed expect to see double-peaked spectra near the edges of the wake (i.e., at the steepest part of the wake shear layer). This occurs because the nearfield of the probe volume senses the inside of the wake while the farfield of the probe volume senses the outside of the wake and/or freestream. This effect is most pronounced when the focal length is longer than our 2.5D example case, but it still can occur at the 2.5D position. This is a limitation of the study in its current form. Related to the good fit of the Machine Learning data in Figure 10(d), these data are all at $\delta=0.2°$, so the lidar beam is not intersecting the wake edges for these cases. One would not expect to see double peaks here, therefore. Figures 16 (a) and (b), on the other hand, show data near the wake edges (i.e., 18

and 45 m), and there is generally a small increase in error for the Machine Learning technique compared to the Advanced Filtering one, though the error bounds are large, as noted in the manuscript.

- I would prefer to have the experiment site description (Sect. 3) before the data processsing description (Sect. 2) as I think it might ease the reading process. Especially it will also make the section about the ML easier to read. Yes, we agree with your point and have made the switch.

- 528: "This source stems from the difference..." Your explanation to this shows plausibility for this interpretation but is not sufficient to exclude all other possible sources which could cause such a bias. For this you would need to deliver a more quantitative estimate of your interpretation. Anyways, it is maybe beyond the scope of this manuscript so you might simply go for "This source PROBABLY stems from the difference..." OK, we have made this addition.

- Fig: 10: The green "x" can be misinterpreted as a fully trusted data point (instead of an outlier well handled by your method) if not carefully reading the legend. Another notation would be preferred. Yes, please see Figure 9 in the new manuscript. We wanted to keep the color green for these data points to help associate the data with the machine learning technique rather than the thresholding technique (which is colored red), so we have replaced the green "x" with a green "o" and then overlaid a red "x" on top.

- 709: Please add a reference to the out-of-distribution-detection techniques OK, please see the added reference to Yang et al., 2021.

- 734: In what you show rather than providing "higher accuracy" ML appears to provide "very similar accuracy" to the advanced filtering method. Please re-word This is a good point, and we have removed the mention of accuracy here altogether. We believe that with future tuning of the machine learning technique, accuracy COULD improve beyond that of the advanced filtering technique, but we agree that hasn't been demonstrated yet.

Technical Corrections
- 331: please include space between "3." and "Experimental Techniques" This has been corrected here and in several other places; thank you.